# Single-cell transcriptome and translatome dual-omics reveals potential mechanisms of human oocyte maturation

Wenqi Hu[1,8], Haitao Zeng[2,8], Yanan Shi[2], Chuanchuan Zhou[2], Jiana Huang[2], Lei Jia[2], Siqi Xu[1], Xiaoyu Feng[1], Yanyan Zeng[2], Tuanlin Xiong[3], Wenze Huang[3], Peng Sun[2], Yajie Chang[2], Tingting Li[2], Cong Fang[2], Keliang Wu[4], Lingbo Cai[5], Wuhua Ni[6], Yan Li[6], Zhiyong Yang[7], Qiangfeng Cliff Zhang[3], RiCheng Chian[7], Zijiang Chen[4], Xiaoyan Liang[2]✉ & Kehkooi Kee[1]✉

The combined use of transcriptome and translatome as indicators of gene expression profiles is usually more accurate than the use of transcriptomes alone, especially in cell types governed by translational regulation, such as mammalian oocytes. Here, we developed a dual-omics methodology that includes both transcriptome and translatome sequencing (T&T-seq) of single-cell oocyte samples, and we used it to characterize the transcriptomes and translatomes during mouse and human oocyte maturation. T&T-seq analysis revealed distinct translational expression patterns between mouse and human oocytes and delineated a sequential gene expression regulation from the cytoplasm to the nucleus during human oocyte maturation. By these means, we also identified a functional role of OOSP2 inducing factor in human oocyte maturation, as human recombinant OOSP2 induced in vitro maturation of human oocytes, which was blocked by anti-OOSP2. Single-oocyte T&T-seq analyses further elucidated that OOSP2 induces specific signaling pathways, including small GTPases, through translational regulation.

Gene expression is often deduced from the transcriptome based on the central dogma of molecular biology. However, post-transcriptional regulation and translational control of gene expression may alter the initial transcriptional profile[1]. Therefore, translatome is more accurate than transcriptome as a measure of ultimate gene expression[2], especially in cells that undergo inactive transcriptional programs or local translational regulation, such as oocytes, neurons, and immune cells[3–5]. Hence, simultaneously measuring transcriptomes and translatomes would provide a more comprehensive and accurate assessment of gene expression.

Understanding the mechanisms underlying oocyte maturation is essential for reproductive biology and medicine. From the germinal vesicle (GV) stage, oocytes reduce their genome-wide transcriptional activities and rely on translational regulation to control gene expression until zygotic activation in early embryos[6]. Oocyte maturation depends not only on nuclear maturation but

[1]Center for Stem Cell Biology and Regenerative Medicine, Department of Basic Medical Sciences, School of Medicine, Tsinghua University, 100084 Beijing, China. [2]Reproductive Medicine Research Center, The Sixth Affiliated Hospital of Sun Yat-sen University, 510655 Guangzhou, China. [3]MOE Key Laboratory of Bioinformatics, Beijing Advanced Innovation Center for Structural Biology and Frontier Research Center for Biological Structure, Center for Synthetic and Systems Biology, School of Life Sciences, Tsinghua University, 100084 Beijing, China. [4]Center for Reproductive Medicine, Cheeloo College of Medicine, Shandong University, 250012 Jinan, Shandong, China. [5]State Key Laboratory of Reproductive Medicine, Center of Clinical Reproductive Medicine, First Affiliated Hospital, Nanjing Medical University, 210029 Nanjing, China. [6]Reproductive Medicine Center, The First Affiliated Hospital of Wenzhou Medical University, 325000 Wenzhou, Zhejiang Province, China. [7]Center for Reproductive Medicine, Shanghai Tenth People's Hospital of Tongji University, 200072 Shanghai, China. [8]These authors contributed equally: Wenqi Hu, Haitao Zeng. ✉e-mail: liangxy2@mail.sysu.edu.cn; kkee@tsinghua.edu.cn

also on cytoplasmic maturation, which is equally important but less investigated[6,7]. Consequently, determining the translational profiles of both cytoplasmic and nuclear genes that regulate oocyte maturation is essential to providing a comprehensive study of oocyte maturation processes.

Recent single-cell multi-omics studies and stem cell in vitro differentiation assays have opened new avenues for studying gene expression in human oocytes. However, these studies have been mainly focused on transcriptional expression[8,9]. Although mass spectrometry proteomics enables the most direct detection of functional genes in cells, limitations still exist for the quantitative measurement and accuracy of low abundant –s at the single-cell level[10]. Recent studies have shown that translatomes can indicate genome-wide translational expression through high-throughput sequencing of mRNA bound by ribosomes (ribosome-protected fragments, RPFs), a technique referred to as ribosome profiling or Ribo-seq[11,12]. Although the single-cell Ribo-seq protocol is quite developed, it still requires cell pooling for gel separation and library construction; therefore, no transcriptome profile can be provided from the same cells[13]. Moreover, the short sequencing reads of Ribo-seq led to a low mapping rate and a much higher sequencing cost. Additionally, RiboTag[14,15] requires tagging of the ribosomal subunit and genomic manipulation, and Ribo-STAMP[16] requires the expression of a ribosomal subunits-APOBEC1 fusion protein to modify their RNA targets. Therefore, these methods are not feasible for human and clinical samples. Fortunately, a recently developed tagging-free method, termed RiboLace, which utilizes puromycin-analog binding to the A site of translating ribosomes, was demonstrated to faithfully profile actively translated ribosomes in cells without genetic manipulations[17].

In this study, we developed a transcriptome and translatome sequencing (T&T-seq) strategy that combines RiboLace for translatomes and SMARTer-seq for transcriptomes using single human oocytes or embryos. Based on T&T-seq, we uncovered the differentially translated genes at different developmental stages in human and mouse oocytes. Furthermore, we applied single-cell T&T-seq to examine the effect of the oocyte secreted protein, OOSP2, on human oocyte gene expression and demonstrated that OOSP2 induces oocyte maturation through the translational upregulation of many downstream genes.

## Results

### Development of T&T-seq for single oocyte and single cell translatome profiling

Translatome profiling of T&T-seq relies on affinity purification of actively translating ribosomes using puromycin analog magnetic beads (RiboLace)[17] which bind to the A-site of translating ribosomes. Ribolace specifically captures actively translating ribosomes in the cellular lysate without the requirement of tagging ribosomal proteins (Fig. 1a). Although the initial protocol was developed for translatome profiling of bulk cells[17], in this study, we have developed a dual-omics protocol for both translatome and transcriptome using low-input and single mouse oocytes (Supplementary Data 1). The lysates were divided into two parts: one for transcriptome analysis and one for translatome profiling. In this sense, one part of the cellular lysate was harvested for regular total RNA sequencing, whereas the remaining lysate was used for RiboLace purification of actively translated RNAs. Independent reverse-transcribed libraries were prepared separately for sequencing, which yielded transcriptome and translatome profiles for the same samples (Fig. 1a). We validated the protocol by running different ratios of cell lysates (20%, 50%, and 100% for transcriptomes and 50%, 80%, and 100% for translatomes), and mouse oocytes (from 10 to 1 oocytes). The results showed high correlation coefficients (0.85–1.00), indicating that the T&T profiles remained highly similar regardless of the different ratios and oocyte numbers (Fig. 1b). Moreover, different inputs of 293FT cells (1–1000) were subjected to T&T-

seq, which confirmed that five single-cell replicates were also highly correlated with the 10–1000 cell samples (Fig. 1c). To test the false positive rate of the translatome in T&T-seq, we incorporated ERCC spike-ins into single-cell lysates before RiboLace purifications and estimated approximately 0.167 of non-specific binding (Supplementary Fig. 1a), which is significantly lower than the reported false positive rate 0.3 of RiboTag[15]. Nevertheless, when we compared our translatome results to the published RiboTag profiles using 200 mouse GV oocytes[14], the correlation coefficients ranged from 0.77 to 0.82, showing a high similarity between the resulting translatomes using the two methods (Supplementary Fig. 1b).

To compare the translatomes using an independent profiling method, we developed a small-scale Ribo-seq (miniRibo-seq) using a combination of RNase T1 and RNase A digestion in 50 oocytes (Supplementary Fig. 1c). The average mapping rates of the T&T-seq translatome for 10 mouse oocytes and single oocytes reached 91.45% and 79.29%, respectively, while that of miniRibo-seq only reached 21.60% (Supplementary Fig. 1d, Supplementary Data 2). MiniRibo-seq showed typical periodicity and mapped to the open reading frames (ORFs) of genes expressed in oocytes (Supplementary Fig. 1e–g), indicating that T&T-seq has higher coverage and is more cost-effective than Ribo-seq. Additionally, principal component analysis (PCA) showed that the translatomes of GV oocytes using T&T-seq, miniRibo-seq, and Ribotag-seq clustered on one side and were well-separated from the cluster of meiosis I (MI) or meiosis II (MII) oocytes (Fig. 1d). These findings indicate that T&T-seq translatomes from single oocytes produced similar profiles to those from miniRibo-seq and Ribotag-seq, in which 50 oocytes or 200 oocytes at GV and MII stages are needed, respectively.

To identify differentially expressed genes (DEGs) during oocyte maturation using T&T-seq, we selected highly translated genes in GV or MII by combining 10 and single oocyte samples (Supplementary Fig. 2a). A total of 464 GV-enriched and 439 MII-enriched genes were identified by comparing their expression at the GV and MII stages. Volcano plots and gene ontology (GO) analysis both confirmed that T&T-seq of 10-oocyte and 1-oocyte samples, together with miniRibo-seq samples, produced similar profiles of DEGs and differentially regulated pathways between GV and MII stages (Fig. 1e, Supplementary Fig. 2b, c). For example, *Nobox, Oosp2, Tubb4b*, and all three members of the ZP family were significantly higher in mouse GV oocytes than in MII oocytes; however, *Btg4, Cnot7, Dazl, H1foo, Obox5*, and *Oosp1* were higher in MII oocytes than in GV oocytes (Supplementary Fig. 3). Many genes regulating ribosome biogenesis, translation, and mitochondrial protein complexes were highly translated in GV mouse oocytes, whereas genes involved in the regulation of the cell cycle, DNA recombination, and chromosome segregation were enriched in MII oocytes (Supplementary Data 3). Hence, the consistency and reproducibility of T&T-seq at the single oocyte level renders its application for human oocyte analysis, which is limited by the low sample number procurement.

### Distinct translatomes in human GV and MII oocytes

After obtaining ethical approval and consent from donors, we carried out T&T-seq on human GV oocytes, MII oocytes, and morula embryos from donors younger than 35 years. Ten or single oocytes and single morula embryos were subjected to T&T-seq in duplicate. High-quality mapped reads were obtained for these samples, with an average of 12,621 transcribed protein-coding genes [transcripts per million (TPM) > 1] and 13,738 translated protein-coding genes (TPM > 1) detected in these samples (Supplementary Data 4). Comparing the translatomes obtained in our study with those of a recent proteomic study using 100 human GV or MII oocytes[18], over 86% of the GV proteins and 83% of the MII proteins were detected in our translatome analysis (Fig. 2a). These findings confirmed that the translatome genes in this study were translated as proteins in the independent proteomic

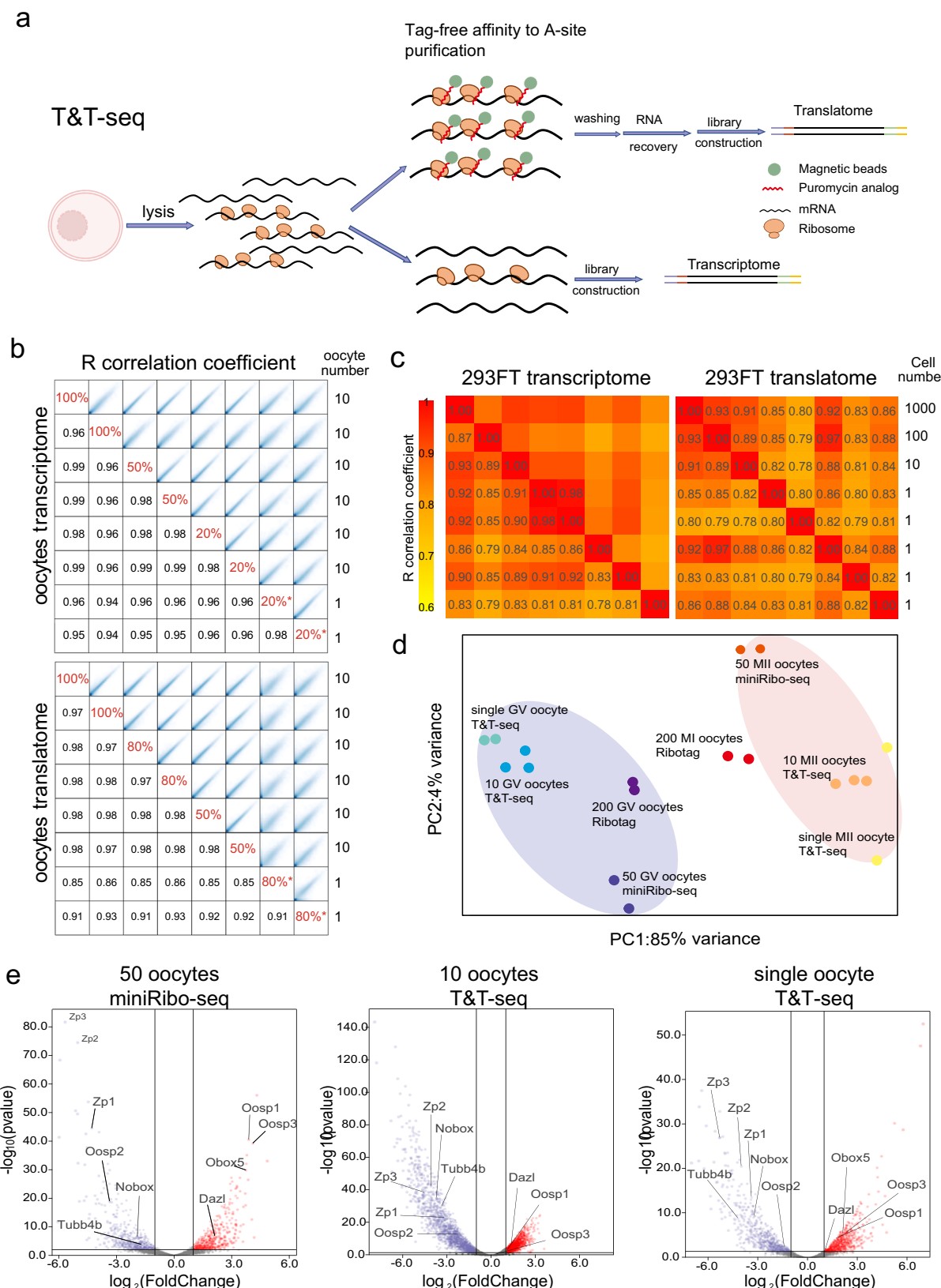

**Fig. 1 | Transcriptomes and translatomes generated by T&T-seq of mouse single-oocytes and 293FT cells. a** Schematic diagram depicting major procedures and principles of T&T-seq. Created with BioRender.com. **b** Correlation coefficient of the transcriptome and translatome data from different oocyte lysate ratios and different oocyte numbers, * indicates single-oocyte samples. **c** Correlation coefficient heatmap of the transcriptomes and translatomes using different number of 293FT cells. **d** PCA plot of T&T-seq and Ribo-tag (published) sequencing data. Blue area covers GV oocytes and the red area covers MII oocytes. **e** Volcano plots showing DEGs identified by miniRibo-seq (50 oocytes) and T&T-seq (10 or single oocyte). $p < 0.05$, wald test, $log_2$ foldchange $> 1$.

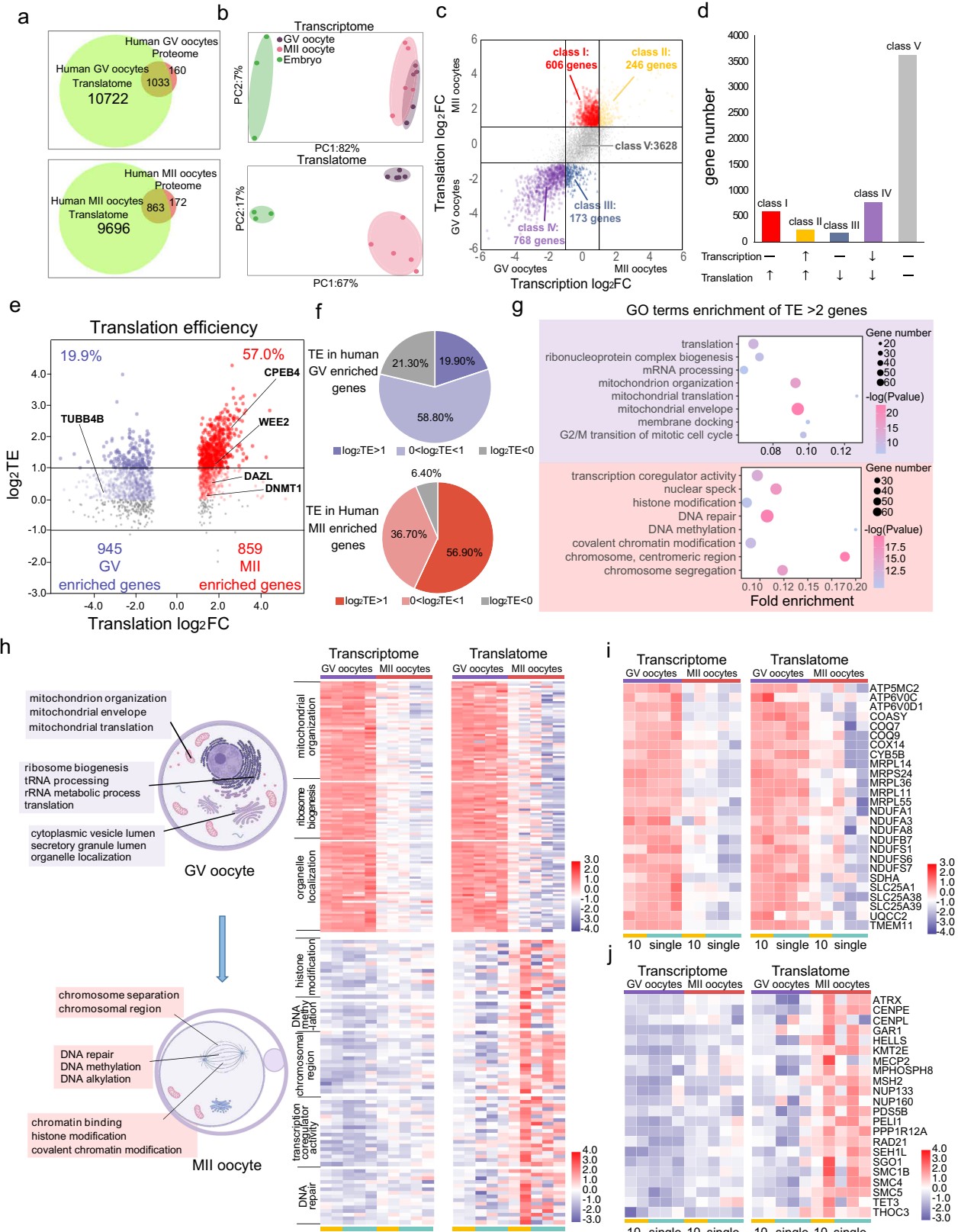

studies. An approximately 10-fold higher number of translated genes (total of 11,756 GV and 10,559 MII genes) were identified with 10–100-fold fewer oocytes in our study compared to those reported by the proteomics approach[18], demonstrating a higher sensitivity and coverage of our T&T-seq, even using much less human-oocyte input material.

Furthermore, PCA of transcriptomes revealed that the morula embryos were clearly separated from the oocytes, whereas GV and MII oocytes partially overlapped with each other (Fig. 2b). In contrast, PCA of the same samples using translatomes separated GV oocytes, MII oocytes, and morula embryos, indicating a more distinct gene expression profile at the translational level than that at the

**Fig. 2 | Distinct transcriptome and translatome patterns of GV and MII human oocytes. a** Venn plot showing the translated gene number using T&T-seq (TPM > 1) and the number of proteins detected in an independent study using 100 oocytes. **b** PCA plots of different development stages of human oocytes and embryos using T&T-seq. Green area covers morula embryos, purple covers GV oocytes, and pink covers MII oocytes. **c** Scatter plot showing the DEGs between GV and MII oocytes in transcriptome and translatome (TPM > 1 in both GV and MII oocytes), p < 0.05, wald test, log2 foldchange > 1. Class I (red) denotes genes translationally upregulated in MII translatome but transcriptionally constant in both stages. Class IV (purple) denotes genes with higher translation and transcription in GV oocytes. **d** Graph showing the number of genes in each class of **c**. Upward arrow indicates upregulated gene expression of either transcription or translation in MII oocytes. The downward arrow indicates downregulated gene expression of either transcription or translation in MII oocytes. Horizontal line indicates gene expression remains unchanged from GV to MII stages. **e** Volcano plot showing the translation efficiency (TE) changes from GV to MII stages. TE of each gene is depicted as $\log_2$TE in GV and MII oocytes. Red color denotes MII enriched genes. Blue color denotes GV enriched genes. Higher intensity of color indicates gene with higher TE. **f** Pie charts showing the proportion of different gene groups of TE of **e**. **g** Representative GO terms enrichment of TE > 2 (or $\log_2$TE > 1) genes. Purple background covers terms enriched in GV oocytes. Red background covers terms enriched in MII oocytes (p-value, hypergeometric test). **h** Representative GO terms with T&T expression heatmaps contrasting highly translated genes in GV and MII oocytes. Genes expression heatmaps of the same sets of genes and the same set of samples with specific GO terms. Created with BioRender.com. **i** Expression heatmap showing specific genes enriched in mitochondrial organization and mitochondrial translation GO terms. **j** Expression heatmap showing specific genes enriched in the chromosomal region and DNA methylation GO terms.

transcriptional level between GV and MII oocytes. This can be mainly attributed to an active translational but inactive transcriptional regulation from GV to MII rather than the gene expression of putative non-coding genes, as we only analyzed protein-coding genes according to the annotated Ensembl database. In addition, the inherent differences between transcription and translation in MII oocytes and the variations among the individual MII oocytes were greater than those in GV oocytes, indicating more variations in MII oocytes than in GV oocytes (Supplementary Fig. 4a–c).

To further understand the genome-wide changes in transcriptomes and translatomes, we analyzed DEGs in GV and MII oocytes at the transcriptome and translatome levels (Fig. 2c). Different classes of genes were categorized according to their expression at the transcriptome and translatome levels and the two oocyte stages. Interestingly, most of the MII-enriched genes had a higher translated expression (>2-fold); however, these were observed to be transcriptionally constant in both stages (class I, 606 genes). In contrast, most of the GV-enriched genes were transcriptionally and translationally higher than those in the MII stages (class IV, 768 genes). Overall, most of the expressed genes remained constant or decreased at the transcriptional level from the GV to MII stages (Class I + III + IV + V) (Fig. 2d), which is consistent with the conventional model of genome-wide inactivation of transcriptional activities during mammalian oocyte maturation. GO analysis revealed that many Class I genes were enriched in the regulation of chromosome organization, meiotic cell cycle, and DNA methylation, suggesting that the expression of genes involved in regulating meiotic chromosomes was translationally increased, without any changes in their transcriptional expression (Supplementary Fig. 5). In contrast, most of the genes translated and transcribed at higher levels in GV than in MII oocytes (Class IV) were identified to be involved in translation and mitochondrial organization. To further investigate the transcriptional abundance of the highly translated genes, we plotted the expression data obtained by T&T-seq for the actively translated genes in GV and MII oocytes (Supplementary Fig. 6), which showed that when the low abundance at the transcriptional level was set at 1 < TPM < 10, there were approximately 71 and 279 low-abundance transcripts in GV and MII oocytes, respectively (Supplementary Data 5).

## Sequential translational regulation in human oocytes

If the translation is more active than the transcription during oocyte maturation from GV to MII oocytes, then more genes are expected to show high translational efficiency (TE). Indeed, 57% of MII-enriched genes had a TE > 2 (or $\log_2$TE > 1), but only 19.9% of GV-enriched genes showed TE > 2 (Fig. 2e, f), clearly indicating more genes with a high TE in MII than in GV oocytes. According to the analysis, genes, such as *CPEB4*, *WEE2*, *DAZL*, and *DNMT1*, had a TE > 1 in MII oocytes, while *TUBB4B* had a TE > 1 in GV oocytes (Fig. 2e). GO terms with TE > 2 during the GV stage included mitochondrial organization, ribosome biogenesis, and organelle localization, whereas GO terms with a TE > 2

during the MII stage included histone modification, DNA methylation, chromosomal region, transcriptional co-regulator activity, and DNA repair (Fig. 2g, Supplementary Data 6).

Consistent with the above analysis, when heatmaps of the highly translated genes at the GV and MII stages were generated, both the transcription and translation levels were higher in the GV stage than in the MII stage. However, the heatmaps of the MII-enriched genes showed a markedly higher translational expression (Fig. 2h). Interestingly, most of the GV-enriched genes were involved in cytoplasmic processes, while MII-enriched genes were mainly involved in nuclear or chromosomal processes, suggesting that translational upregulation occurs sequentially from the cytoplasm to the nucleus during GV to MII maturation. For instance, GV-enriched genes included many mitochondria-related genes involved in the assembly of the oxidative phosphorylation system (OXPHOS)[19], such as *COX14*, *NDUFS1*, and *UQCC2* (Fig. 2i), suggesting that the assembly of OXPHOS mainly occurs at the GV stage. In contrast, genes involved in DNA modifications and chromosomal regulation, such as *ATRX*, *CENPL*, and *SMC5*, were translated at MII stage, following the need for meiotic I and II chromosomal segregations (Fig. 2j). Moreover, the expression levels and patterns were highly similar between 10-oocyte and single-oocyte samples for both transcriptome and translatome analyses, indicating the consistency and reproducibility of single-oocyte T&T-seq.

## Specific binding motifs at 3′-untranslated region (UTR) of highly translated genes

Previous studies in mice revealed that a high translational expression during oocyte maturation is regulated by translational regulatory elements bound by RNA-binding proteins (RBPs) to the 3′-UTR of the genes[3,20,21]. However, the identification of RBPs and enriched motifs in human oocytes have not yet been reported because of a lack of translatome datasets. Therefore, we screened the 3′-UTR of high-TE genes (TE > 1) in human GV and MII oocytes to identify potential RBP sites and thus investigate RBPs as potential translational regulators. A total of 112 and 160 RBPs, acting as potential translational regulators, were identified in the high-TE genes of GV and MII oocytes, respectively (Fig. 3a, Supplementary Data 7). We focused on analyzing RBPs and their motifs at MII stage, which most likely functions during oocyte maturation. The binding motifs of *FXR1*, *PCBP1*, *DDX3X*, *CPEB2*, and *CPEB4* were the top five motifs among the high-TE genes in MII and were expressed in the oocytes examined (Fig. 3b, c). Homologs of *FXR1*, *PCBP1*, *CPEB* family genes, and *DDX3X* have been reported to participate in translation regulation in animal oocytes[20,22–24], supporting their potential regulatory roles during human oocyte maturation.

Next, we analyzed the 3′-UTR of the MII high-TE genes to determine whether there are commonly regulated genes that carry multiple RBP motifs within the top five RBPs. Interestingly, 63 genes carried all five RBP motifs at their 3′-UTR, including *AGO2*, *CNOT6*, and *DDX17* (Fig. 3d, e). GO analysis of the 63 genes revealed that they participate in the transcriptional co-regulatory activity, TOR signaling, and mRNA

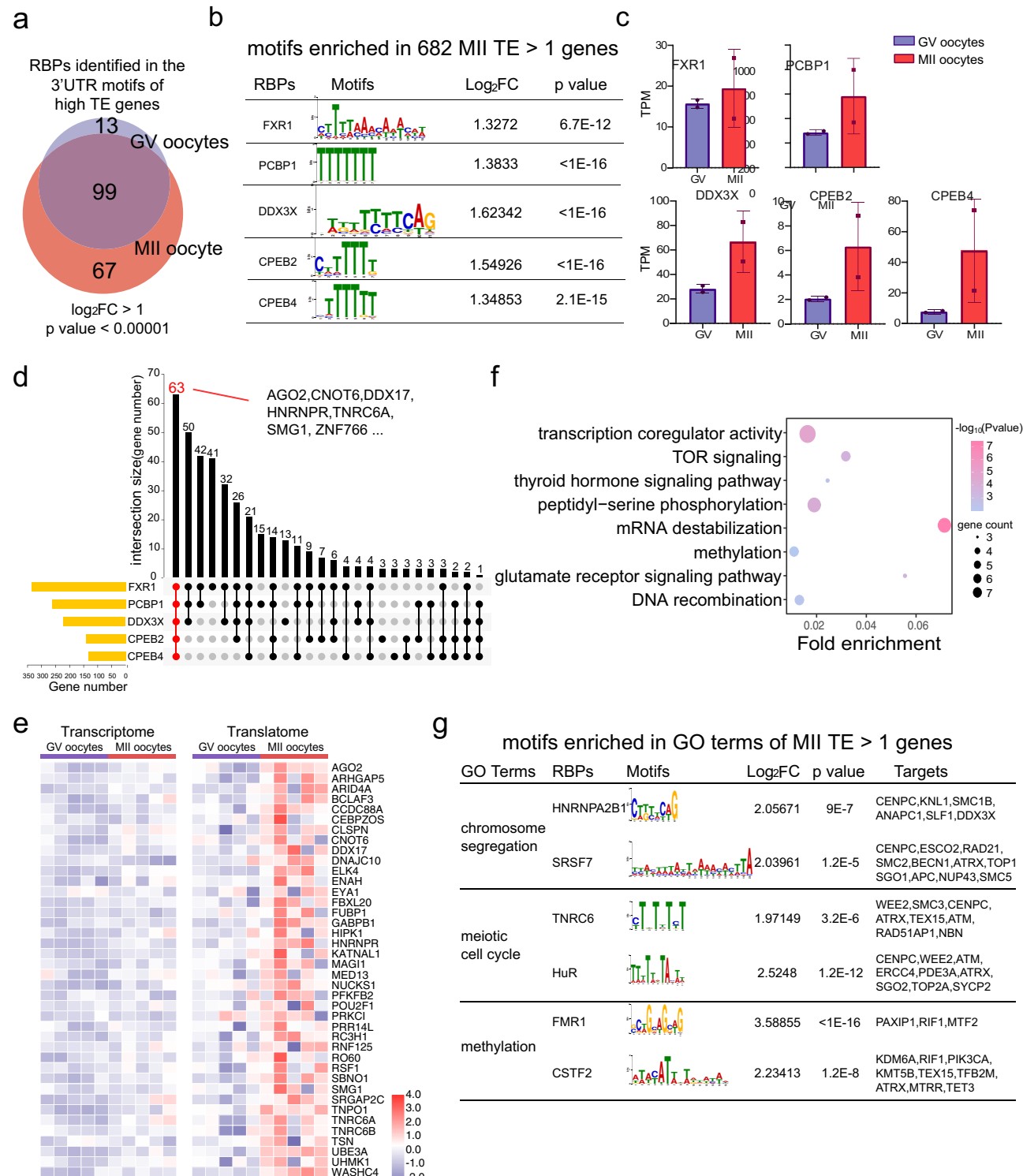

**Fig. 3 | High TE genes carry RNA-binding-protein motifs at their 3'UTRs with high frequency. a** Venn plot of putative RBPs identified in 3'UTR motifs of high TE genes in GV and MII oocytes, log2Fold change > 1, $p$ < 0.00001, one-sided $t$-test (see the "Methods" section). Blue circle covers RBPs bound to 3'UTR of high TE in GV oocytes. Red circle covers RBPs bound to 3'UTR of high TE in MII oocytes. Intercepted area indicates RBPs bound to 3'UTR of high TE in both GV and MII. **b** 5 RBPs with the high significant values and their binding motifs enriched in MII TE > 1 gene ($p$-value, one-sided $t$-test). **c** The translational expression levels in GV and MII of the top 5 RBPs shown in **b**. Data from two biological replicates of 10 human oocytes. Error bar denotes standard deviation (SD). **d** Upset plot showing the number of the overlapped targeted genes containing the five RBP motifs. 63 genes including AGO2, CNOT6 are potentially co-regulated by the 5 RBPs. **e** T&T heatmap of representative genes potentially co-regulated by 5 RBPs. **f** Representative GO terms of the genes co-regulated by the 5 RBPs ($p$-value, hypergeometric test). **g** motifs enriched in GO terms of MII TE > 1 genes ($p$-value, one-sided $t$-test).

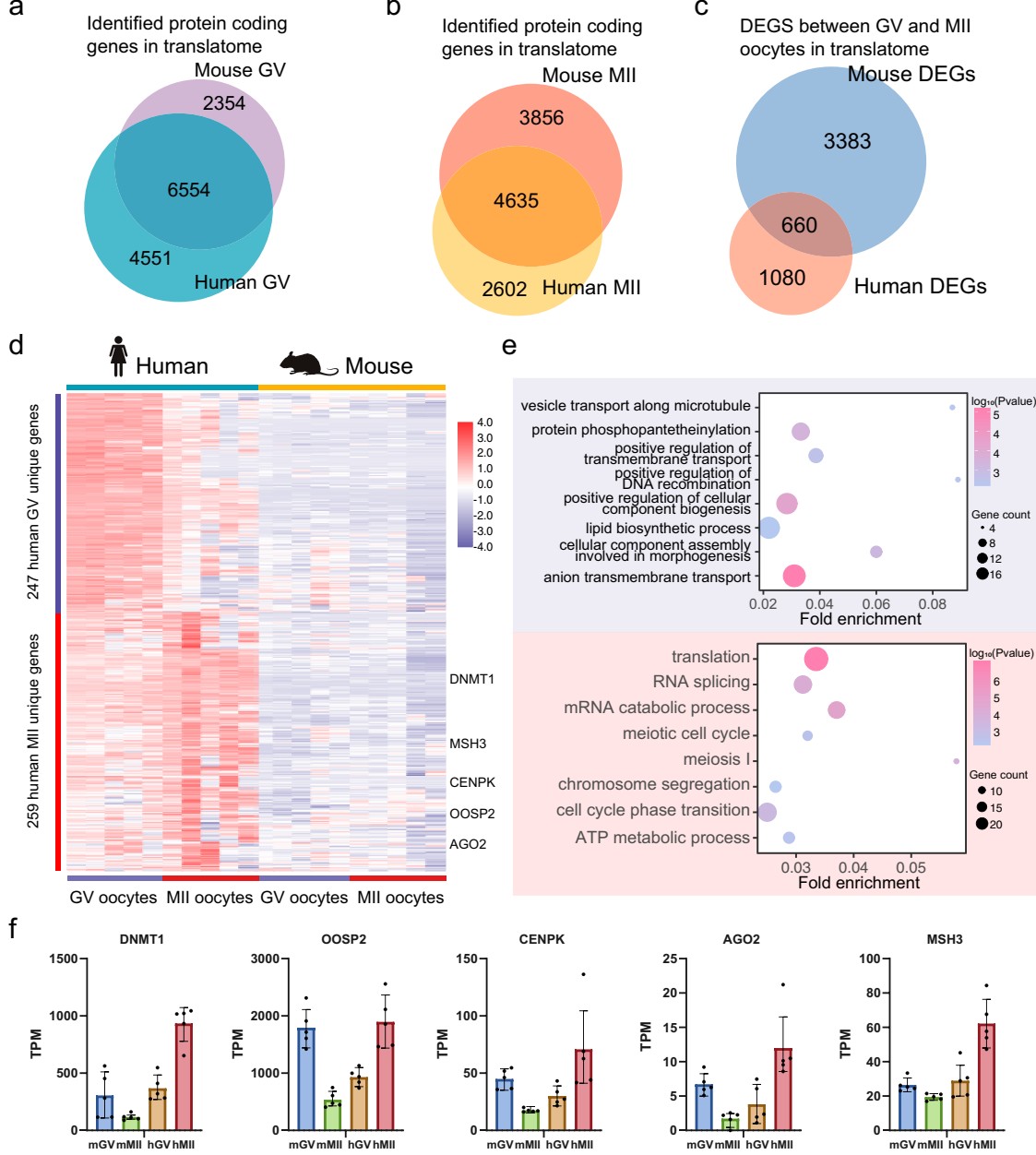

**Fig. 4 | Translatome difference between human and mouse. a** Veen plot shows overlap of translatome of GV oocytes between human and mouse. **b** Veen plot shows the overlap of translatome of GV oocytes between human and mouse. **c** Veen plot shows the overlap of DEGs identified by translatome between human and mouse. **d** Expression heatmap of human unique DEGs. **e** GO enrichment of human unique DEGs. Blue background shows terms enriched in human GV oocytes. Red background shows terms enriched in human GV oocytes. Specific human MII enriched genes are labeled on the right (*p*-value, hypergeometric test). Created with BioRender.com. **f** Translational expression levels of the representative genes of human and mouse GV and MII oocytes. Data from five biological replicates of human and mouse oocytes. Data are presented as mean values ± SD.

destabilization, among other functions (Fig. 3f), suggesting that these genes are simultaneously regulated by the top five RBPs in the same cellular pathway. Moreover, we selected high-TE genes enriched in GO terms related to oocyte maturation, such as "chromosome segregation", "meiotic cell cycle", and "methylation", observing that motifs and gene targets were predicted using their 3′-UTR sequences (Fig. 3g).

**Uniquely translated genes in human versus mouse oocytes**

Mouse models are often used to study and infer the mechanisms of human oocyte maturation; however, unique translational expression and regulation may exist in human oocytes. Based on the T&T-seq of mouse and human oocytes, we compared the translatomes of the two

species. There were 6554 (59% of human GV translatome) and 4635 (64% of human MII translatome) commonly translated genes in GV and MII oocytes of humans and mice, respectively. However, only 660 DEGs (38% human) of GV versus MII were found between the two species (Fig. 4a–c). These findings suggest that the translational regulation of DEGs is less conserved than the translational expression between the two species. Interestingly, some common DEGs showed the same expression trend from GV to MII oocytes in both species (Supplementary Fig. 7a). For instance, *MRPL36*, *GEMIN6*, and *TUBB4B* were translated more in GV than in MII oocytes, whereas *DNMT3B*, *BRCA2*, *MSH2*, and *SYCP2* were translated more in MII than in GV oocytes in both species (Supplementary Fig. 7b).

Next, by selecting highly translated DEGs in human oocytes but relatively less translated in mouse oocytes, we analyzed unique human DEGs. Results yielded 247 GV-specific and 259 MII-specific genes (Fig. 4d, Supplementary Data 8). Regarding the GV-specific DEGs, GO analysis indicated the genes enriched in pathways, including "lipid biosynthesis", "anion transmembrane transport", and "positive regulation of cellular component biogenesis" (Fig. 4e). More interestingly, the human MII-specific genes included GO terms for "translation", "meiosis I", and "chromosome segregation" (Supplementary Data 9), suggesting that these processes are differentially regulated between the two species using different sets of genes through translational regulation. Among these unique human MII genes, *DNMT1*, *MSH3*, *CENPK*, *OOSP2*, and *AGO2* showed opposite trends in translational expression in humans versus mice (Fig. 4f). For instance, *DNMT1* was translationally downregulated from GV to MII oocytes in mice, while it was upregulated by more than 2-fold from GV to MII oocytes in humans. Finally, we further identified the top ten, mouse-specific RBPs that may regulate highly translated genes in MII oocytes based on the enrichment of 3′-UTR motifs as shown in Supplementary Fig. 8 and Supplementary Data 10.

## Cytoplasmic polyadenylation element (CPE) motif in the 3′-UTR regulates the oocyte secreted proteins (OOSPs)

Strikingly, the existence of CPE and polyadenylation signals (PAS) in the 3′-UTR of the OOSP family genes strongly correlates with their translational expression based on T&T-seq. Mouse *OOSP2* has only one PAS at the 3′-UTR, and it is translated at lower levels than *OOSP1* and *OOSP3*, which carry CPE and a neighboring PAS (Fig. 5a). In contrast, human *OOSP2* is the only *OOSP* gene carrying both CPE and PAS and is the only OOSP protein highly translated in human oocytes. Based on the CPE motifs in human *OOSP2*, the translational regulator might be a CPEB protein. Translatome analysis indicated that CPEB4 expression was higher than that of CPEB2 in human MII oocytes (Fig. 5b). Therefore, we tested whether OOSP2 3′-UTR is regulated by CPEB4 by conducting a dual-luciferase assay. As expected, when CPEB4 was overexpressed in 293FT cells, cells transfected with a luciferase reporter plasmid containing the OOSP2 3′-UTR showed significantly higher luciferase activity (Fig. 5c). Furthermore, when both CPEs were mutated at the OOSP2 3′-UTR, reduced levels of luciferase were detected. Taken together, these results suggest that CPEB4 binds to the OOSP2 3′-UTR and upregulates its translational expression. Moreover, we found that the top nine secreted proteins (Fig. 5d, Supplementary Data 11) contained at least one CPE next to one or more PAS within 100 nucleotides (Fig. 5e), suggesting that the top secreted proteins are translationally regulated during human oocyte maturation[14].

## OOSP2 promotes human oocyte maturation in vitro

Protein–protein interaction (PPI) analysis by String[25] showed that, unlike mouse OOSP2, human OOSP2 has a co-expression relationship with BMP15 and H1FOO (Supplementary Fig. 9a, b), suggesting that OOSP2 may participate in folliculogenesis and oocyte maturation. We further confirmed that OOSP2 was co-expressed with BMP15 and localized to human oocytes by immunostaining of human ovarian sections (Supplementary Fig. 9c). If OOSP2 is secreted by human oocytes to induce their maturation, the addition of human recombinant OOSP2 (hRec-OOSP2) to in vitro maturation (IVM) culture of GV oocytes may increase the percentage of mature oocytes. To test this possibility, we collected GV oocytes from consent donors and used an incubator equipped with a time-lapse microscope to monitor oocyte maturation. GV breakdown (GVBD) and polar body extrusion (PBE) are common morphological changes that follow two stages during oocyte maturation (Fig. 6a); hence, the latter was used to identify mature oocytes at the MII stage. Pairwise GV oocytes from the same donor were compared for each set of experiments to eliminate the

heterogeneity of different donors. Of the 24 pairs of GV oocytes, 21 oocytes incubated with hRec-OOSP2 matured (87.5%) versus 15 in the control group (67%) (Fig. 6b), indicating a higher maturation rate in the group incubated with hRec-OOSP2.

If OOSP2 functions as a secreted protein to induce oocyte maturation, the addition of an antibody against OOSP2 during IVM culture may block or slow down oocyte maturation. To test this hypothesis, 10 of the 16 GV oocytes incubated with the OOSP2 antibody (α-OOSP2) did not mature (62.5%). However, only 4 of the 16 GV (25%) oocytes stopped maturation in the control group (Fig. 6c, d). This finding indicated that incubation of GV oocytes with an OOSP2 blocking antibody impairs oocyte maturation rate. Altogether, the blocking antibody and the recombinant OOSP2 experimental results support the role of OOSP2 as an oocyte maturation factor, even though some oocytes incubated with hRec-OOSP2 reached GVBD and PBE faster (approximately 2 h) than that observed for control oocytes (Fig. 6a, e), and some oocytes incubated with α-OOSP2 showed slower GVBD and PBE (approximately 5 h) than that observed for control oocytes (Fig. 6c, f).

In addition, using an additional 12 pairs of GV oocytes (each pair was donated by the same donors) and an independent knockdown approach (Trim-Away experiment[26]), we observed that the maturation rate of human oocytes was also reduced and delayed, a finding consistent with the results obtained using the antibody blocking approach (Supplementary Fig. 10a, b). Moreover, we analyzed the translatome changes after Trim-Away treatment with OOSP2 protein in human oocytes using T&T-seq. We found that genes, such as *ZP3*, *POMZP3*, and *GDF9*, were downregulated after OOSP2 Trim-Away treatment (Supplementary Fig. 11a). Additionally, GO analysis showed that the downregulated genes were enriched in "organelle localization", "female gamete generation", and "chromosome segregation", which are related to oocyte maturation, suggesting that OOSP2 disrupts oocyte maturation processes (Supplementary Fig. 11b).

## T&T-seq reveals upregulated translation by OOSP2

To elucidate the molecular mechanism of OOSP2 induction, we examined the transcriptome and translatome of human GV oocytes incubated for 1.5 h with hRec-OOSP2, α-OOSP2, or without any additive (all oocytes still maintained GV morphology) to capture the immediate effectors of OOSP2 and thus identify the potentially induced genes or pathways downstream of OOSP2. Using pseudotime analysis, we plotted developmental stages according to the overall gene expression pattern of a cell, as previously reported[27]. Consequently, we observed that the translatomes of the individual α-OOSP2-treated oocytes completely clustered with the control oocytes, whereas the hRec-OOSP2-treated oocytes clustered away from both groups along the trajectory toward either MII oocytes or morula embryos (Fig. 7a). This result suggests that GV oocytes incubated with α-OOSP2 remained immature, while the GV hRec-OOSP2-treated oocytes became more mature although not reaching the MII stage.

To further characterize the induced genes and potential regulatory pathways downstream of OOSP2, we analyzed the T&T changes between control and OOSP2 oocytes (Supplementary Data 12). Differential expression analysis by volcano plots revealed 1649 upregulated genes in the transcriptome and 1225 upregulated genes in the translatome (Fig. 7b, c) between the control and hRec-OOSP2-treated oocytes after 1.5 h of incubation. Although the number of upregulated genes was similar between the transcriptome and translatome after hRec-OOSP2 incubation, there were many more high-TE genes in the hRec-OOSP2-treated group than in the control group (Fig. 7d). Genes with a TE > 4 reached 46.9% versus 3.0% in control oocytes, indicating a strong upregulating effect of OOSP2 on translational expression.

Analysis of the 3′-UTR of the highly upregulated genes (TE > 4) for the RBP motif, resulted in a list of RBPs (top 10-fold change), including *SRSF2*, *SRSF3*, *PTBP1*, *HNRNPC*, *GNL3*, and *HNRNPH3* (Fig. 7e). Further

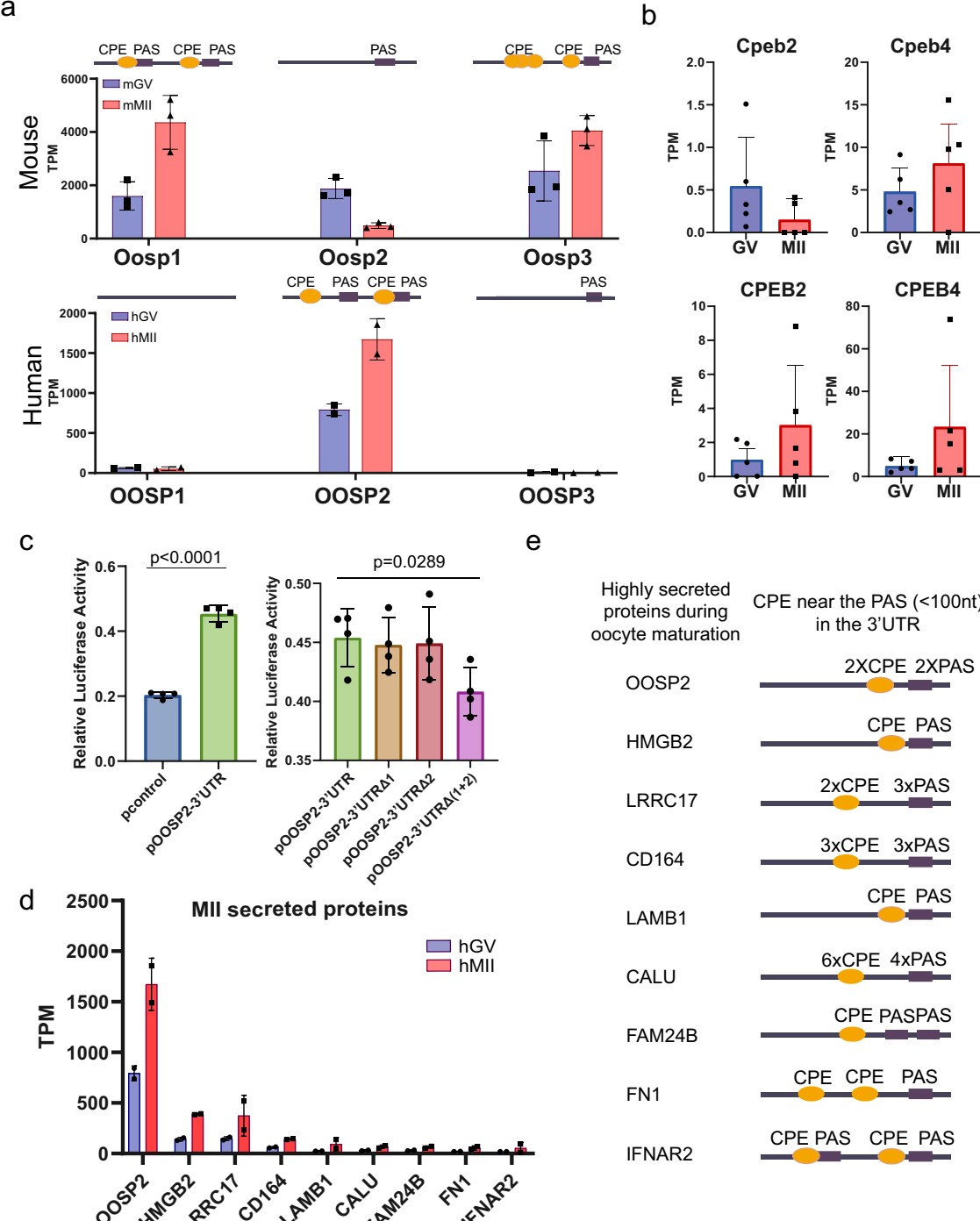

**Fig. 5 | CPE motif in the 3'UTR regulate the oocyte secreted proteins. a** The translational expressions of OOSP gene family in human and mouse GV and MII oocytes. The CPE and PAS elements and positions at the 3'UTRs are indicated on the top of each mouse and human OOSP gene. are indicated on the right of mouse and human OOSP genes. Data from 2 biological replicates of 10 human oocytes and 3 biological replicates of 10 mouse oocytes. Error bar denotes SD. **b** Translational expressions of CPEB2 and CPEB4 $n = 5$ (including 10 and single oocyte data). Data are presented as mean values ± SD. **c** 3'-UTR Luciferase reporter assays of CPEB4 transfected with wild-type or the two mutated CPEB4 elements at the 3'-UTR of OOSP2. $n = 4$, (four independent biological replicates per 293FT cell populations, ~25,000 cells in each sample, Unpaired two-sided $t$-test). Data are presented as mean values ± SD. **d** Translation expression levels of human MII enriched secreted proteins at GV and MII stages. Data from two biological replicates of 10 human oocytes. Error bar denotes SD. **e** 3'UTRs of the MII highly translated and secreted proteins showing the number and approximate location of CPE and PAS motifs.

GO analysis of the highly upregulated genes revealed enriched terms, such as "small GTPase-mediated signal transduction", "response to peptide hormone", and "Ras protein signal transduction" (Fig. 7f). T&T heatmaps of genes enriched in "small GTPase signaling pathway" (Fig. 7g) and "organelle localization" (Fig. 7h) confirmed that the transcriptional levels remained similar between control oocytes and hRec-OOSP2-treated oocytes. However, the translational levels were sharply upregulated in hRec-OOSP2-treated oocytes. Further, network analysis of small GTPase genes and organelle localization genes (Supplementary Fig. 12a, b) showed that many of these genes interact

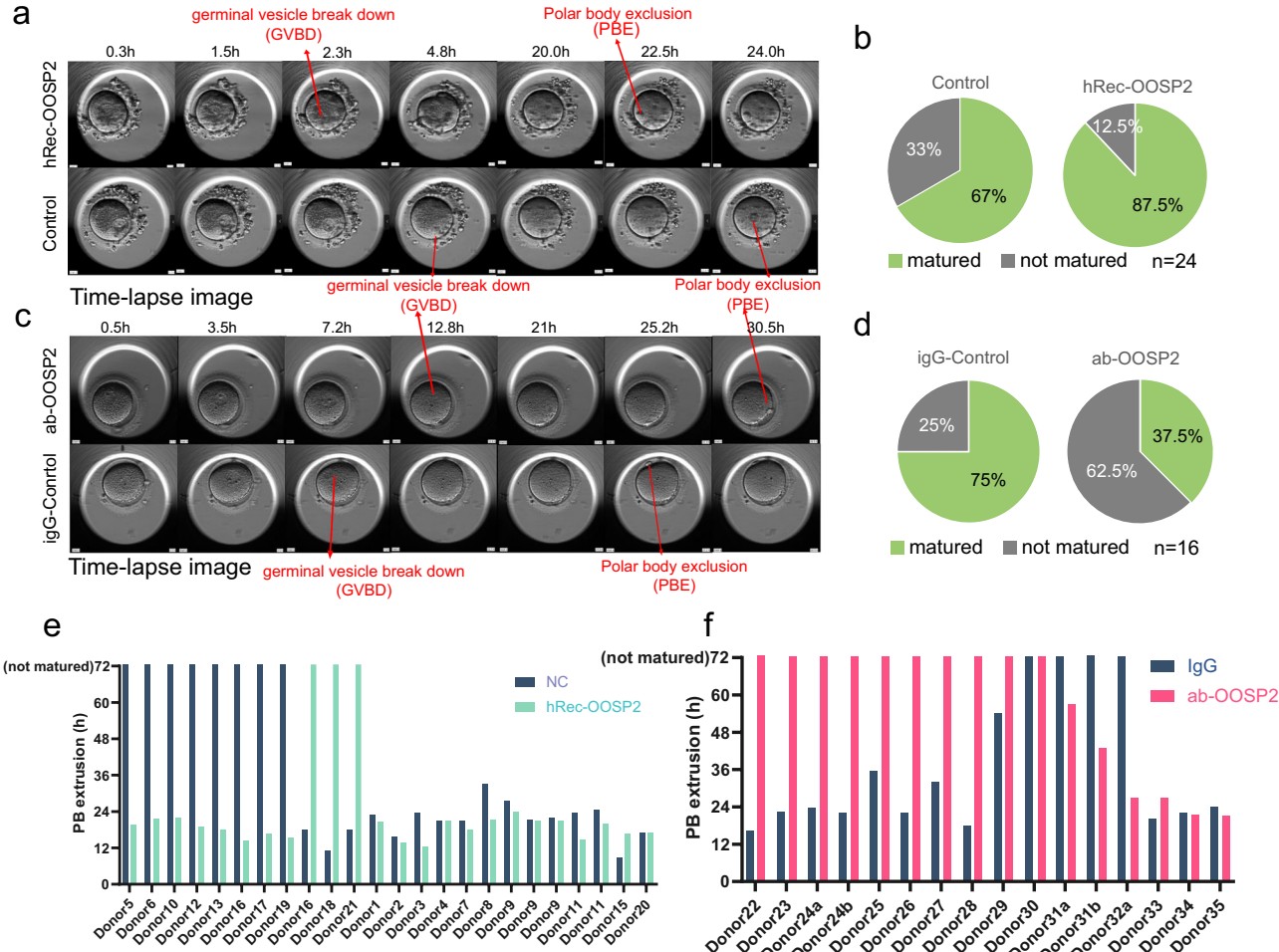

**Fig. 6 | OOSP2 promote human oocytes maturation in vitro. a** Time-lapse image showing the progressive maturation of human GV oocytes in hRec-OOSP2 containing medium (top row) and control medium (bottom row) with indicated time (h). Red arrows indicate the occurrence of GVBD or PBE. **b** Pie chart showing the maturation rate of hRec-OOSP2 treated oocytes and control oocytes. $n = 24$ (21 pairs of independent donors, each pair of oocytes was from the same donor. Chi-square = 2.9484, $p = 0.08586$, Chi-square test). **c** Time-lapse images showing the progressive maturation of human GV oocytes in antibody against OOSP2 (ab-

OOSP2) containing medium (top row) and control medium (ig-Control, bottom row) with indicated time (h). Red arrows indicate the occurrence of GVBD or PBE. **d** Pie chart shows the maturation rate of ab-OOSP2 treated oocytes and igG control oocytes. $n = 16$ (14 pairs of independent donors, each pair of oocytes was from the same donor. Chi-square = 4.57143, $p = 0.03251$, Chi-square test). **e** PBE duration of each pair of oocytes in hRec-OOSP2 and control oocytes. PBE duration over 72 h is counted as not matured. **f** PBE duration of each pair of oocytes in ab-OOSP2 and the control oocytes. PBE duration over 72 h is counted as not matured.

directly or indirectly. Taken together, these results indicate that many genes were simultaneously upregulated with high translational expression without significant transcriptional upregulation after the addition of recombinant OOSP2. Moreover, these genes belong to a group that may induce a cascade of events that regulate oocyte maturation, such as the small GTPase signaling pathway.

## Discussion

In the present study, we demonstrated that T&T-seq can be used at the single-cell level to profile transcriptomes and translatomes of the mouse or human oocytes and embryos. Compared with Ribo-seq and RiboTsag-seq, T&T-seq produced similar translatome profiles, covering approximately 10,000 genes in both mouse and human oocytes. T&T-seq only requires single-oocyte RNA input for dual-omics analyses, while single-cell Ribo-seq requires hundreds of sorted cells pooled together to obtain enough fragments for polyacrylamide gel electrophoresis (PAGE) purification. Therefore, scRibo-seq is suitable for samples with high cell numbers, whereas T&T-seq is more suitable for the ones with low cell numbers, such as oocytes or embryos. Furthermore, the mapping rate of translatomes conducted using T&T-seq was 80–90% compared to approximately 22% using Ribo-seq. This is

because the short-protected transcripts collected in Ribo-seq are often filtered away due to the indistinguishably repeated sequences found in the genomes. Hence, T&T-seq is more cost-effective because less sequencing is required to obtain the same level of mapped sequences as in Ribo-seq. Additionally, both RiboTag and Ribo-STAMP require a fusion tag or fusion protein on the subunit of ribosomes, making them only feasible in animal models or cell lines (Table 1).

The coverage of gene expression in the T&T-seq translatome is markedly higher than the currently used proteomics approaches, which reported approximately 1100 proteins per 100-oocyte sample or approximately 450 proteins per single-oocyte sample[18]. Moreover, quantifying the differential expression of proteins requires isotope labeling in classic proteomics, which limits their usage and accuracy at the single-cell level. On the contrary, T&T-seq allows the measurement of gene expression at both the transcriptional and translational levels, thereby providing a comprehensive recording of both levels of gene regulation in the same genes and the same cells.

Noticeably, the translatomes of human GV and MII oocytes were well separated in PCA, while the transcriptomes partially overlapped, highlighting the translatome as a more accurate reflection of the changes in cellular status and/or developmental stages, such as GV

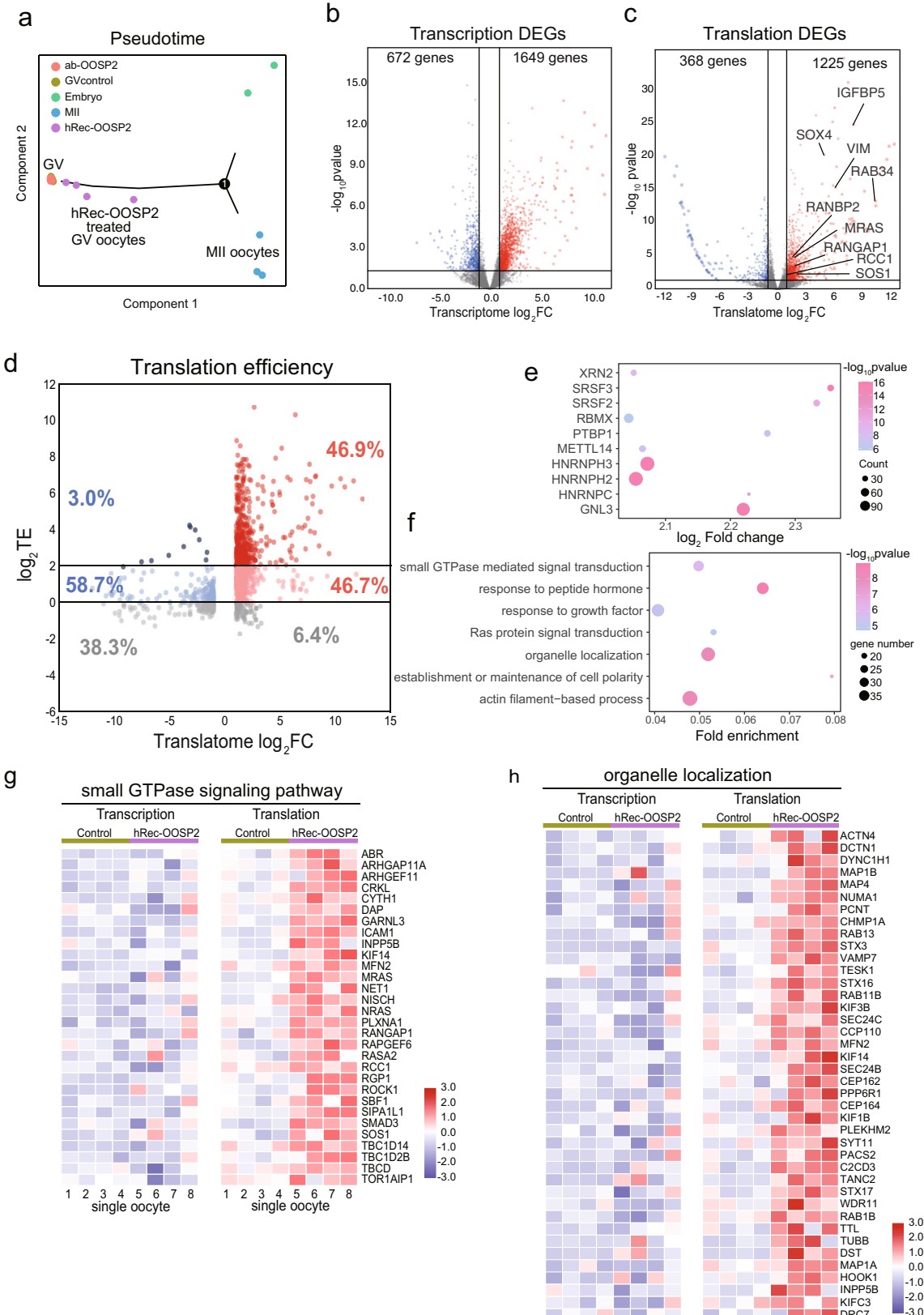

versus MII. Further analysis indicated that the majority of transcribed genes remained unchanged, although more genes were differentially translated from GV to MII oocytes. Therefore, we validated consistent patterns of gene expression in human oocytes with the expected transcriptionally silent and translationally active trends, similarly to the oocytes of other mammalian species[28,29].

A second interesting feature revealed by T&T-seq is the difference in the general expression patterns of highly translated genes between human GV and MII oocytes. The highly translated genes are also highly transcribed in GV oocytes, while in MII oocytes remain mostly unchanged at the transcriptional level. In addition, the high-TE genes in GV were mostly cytoplasmic, while in MII the high-TE genes were

**Fig. 7 | Single-cell T&T-seq helps to reveal transcriptome and translatome changes after OOSP2 treated human oocytes. a** Pseudotime analysis showing the developmental stages of GV control oocytes, hRec-OOSP2 treated oocyte, ab-OOSP2 treated oocytes, MII oocytes, and morula embryos. **b** Volcano plot showing transcriptome DEGs of GV control oocytes versus hRec-OOSP2 treated oocyte (*p*-value, wald test) **c** Volcano plot showing translatome DEGs of GV control oocytes versus hRec-OOSP2 treated oocyte (*p*-value, wald test). **d** Volcano plot showing the TE changes after treated with hRec-OOSP2. TE of each gene is depicted as log2TE of control and hRec-OOSP2 treated oocytes. Red dot indicates upregulated genes in hRec-OOSP2 treated oocytes. Blue dot indicates GV enriched genes. The darker the color, the higher the TE. **e** Putative RBPs (top10 fold change) identified by 3'UTR motifs of TE > 4 genes after treated with hRec-OOSP2 (*p*-value, one-sided *t*-test). **f** Representative GO terms of TE > 4 genes after treated with hRec-OOSP2 (*p*-value, hypergeometric test). **g**, **h** T&T expression heatmap of the OOSP2 upregulated translational genes involved in small GTPase signaling pathway and organelle localization.

**Table 1 | Comparison of T&T-seq with other translatome sequencing techniques**

| | Sequencing reads | Mapping rate | Cell input | Gene detected | Tags or fusion protein | Application | Reference |
|---|---|---|---|---|---|---|---|
| Ribo-seq | RPFs | Low | 10 million cells | >10,000 | No | Large scale sample | Ingolia et al.[57] |
| LiRibo-seq | RPFs | Low | ~5000 cells | >10,000 | No | Small scale sample | Zhang et al.[58] |
| miniRibo-seq | RPFs | Low | ~50 oocytes | >10,000 | No | Small scale sample | This study |
| scRibo-seq | RPFs | Low | Single-cell pool | ~5000 | No | Heterogeneous tissue | Van Insberghe et al.[13] |
| RiboLite | RPFs | Low | Single oocyte | >10,000 | No | Small scale sample | Xiong et al.[59] |
| RiboTag | Full-length | High | 1~100 thousand cells | >10,000 | Yes | Transgenic mouse | Luong et al.[14] |
| Ribo-STAMP | Full-length | High | Single-cell | >10,000 | Yes | Transgenic Cell line | Brannan et al.[16] |
| T&T-seq | Full-length | High | Single oocyte/Single-cell | >10,000 | No | Scarce sample | This study |

involved in nuclear or chromosomal regulation. This finding suggests that cytoplasmic and nuclear genes are not simultaneously upregulated and are more likely to be sequentially upregulated with cytoplasmic genes first followed by the nuclear genes. This sequential pattern from the cytoplasm to the nucleus supports sequential events based on morphological changes during oocyte maturation[6]. Hence, our analyses provide molecular evidence of these morphological changes. Approximately 6.5 h are required for IVM culture of human GV oocytes to reach the GVBD stage, and an additional ~14 h are required to reach PBE[30]. Hence, the highly translated cytoplasmic genes found in GV may suggest that oocytes begin by preparing cytoplasmic machinery and organelles for GVBD, and then later modulate nuclear and chromosomal events to achieve MII oocyte maturation.

Another intriguing finding of this study was the differential expression pattern of mitochondrial genes. Many nuclear genes involved in the assembly of the OXPHOS were highly expressed in GV oocytes, while genes located in the mitochondrial genome were upregulated later. This finding suggests that the assembly of OXPHOS in oocytes is sequentially regulated during maturation and that energy metabolism may be more active in the MII stage when the complexes are fully assembled[19,31]. Based on these findings, our study provides comprehensive information from a global gene expression perspective, covering many pathways and their sequential organization during oocyte maturation.

Cross-species conservation of gene expression between mouse and human oocytes was also investigated using T&T-seq. Interestingly, our analysis indicated that both species conserved the expression of many genes involved in oocyte maturation. However, translated genes were much less conserved, suggesting the use of different sets of genes through translational regulation instead of transcriptional control between the two species. Therefore, functional studies based on gain- and loss-of-function genetic mouse models during oocyte maturation may not be accurate for the identification of DEGs in humans. However, our translatome datasets may serve as an important reference before launching a mouse model-based study of oocyte maturation. One of the most striking examples discovered herein is the differential translational control of *OOSP* family genes. All three members of the family are expressed in mouse oocytes, but only *OOSP2* is highly expressed in human oocytes which correlates with the existence of translational elements at their 3′-UTR. Triple-gene deletion of *OOSP1–3* in mice retains fertility but causes significantly lower blastocyst and reduced offspring number[32], indicating that some oocytes and embryos might be affected without the expression of *OOSP* family genes. However, as the translational regulation and trends of gene expression are distinct in many genes between humans and mice, the requirement of OOSP2 during oocyte maturation and embryo development is expected to be different.

OOSP2 was also the most translated gene among the secreted protein candidates in our datasets. Our experimental results indicated that OOSP2 protein may serve as an autocrine factor to induce oocyte maturation. More importantly, many signaling pathways in GV oocytes were instantly activated by hRecOOSP2 treatment through the upregulation of their translational expression, further strengthening the notion that OOSP2 modulates human oocyte maturation. Among these maturation pathways, many genes are interconnected and related to each other as a cascade of molecular events, as shown in previous reports that studied genes involved in small GTPase signaling pathways[33–36]. Hence, we propose that OOSP2 induces oocyte maturation by upregulating the small GTPase signaling pathway, thereby initiating a cascade of molecular and cellular events from the cytoplasm towards the nucleus.

Many upregulated genes have high-TEs, and their transcriptional levels remain similar after the addition of OOSP2, strongly suggesting that OOSP2 can immediately upregulate the protein levels of these genes without increasing their transcriptional expression. Some examples of the direct regulation of translation without transcription have been reported in neurons[4,37], but not in oocytes. Understanding the detailed mechanism of such occurrences via future studies and determining whether direct translational regulation of gene expression or similar local translational regulation also exists in oocytes and other cell types will be of great interest.

## Methods

### Animal care and use
C57BL/6J mice were purchased from the Vital River Laboratory Animal Technology Co., Ltd. (Beijing, China). In all experiments, 4-week-old female mice were used. All animal care and experimental procedures were performed following the guidelines of the Institutional Animal Care and Use Committee (IACUC) of Tsinghua University, Beijing, China. The mouse facility maintains 12 h/12 h of dark/light cycle, 20–26 °C, and 40–70% humidity.

### Cell lines
293FT cells (Invitrogen) were maintained in 90% Dulbecco's modified Eagle medium, 10% fetal bovine serum, 1 mM GlutaMAX, 0.1 mM MEM

non-essential amino acids, 1 mM sodium pyruvate, 0.5 mg/mL geneticin, and 100 U/mL penicillin–streptomycin. 293FT cells were cultured at 37 °C in a humid atmosphere with 5% $CO_2$.

## Mouse oocytes collection

C57BL/6 female mice were used to collect oocytes. Mice were ethically euthanized, and ovaries were retrieved and immediately placed into M2 solution pre-heated at 37 °C on a warming stage attached to a microscope. GV oocytes were collected from ovarian follicles by needle pricking in the presence of hyaluronidase (ART-4007-A, SAGE) to remove the cumulus cells. For the collection of MII oocytes, 11 IU PMSG (220 μL) was injected into the abdomen for 47.5 h. The mice were then stimulated with 11 IU of human chorionic gonadotropin (hCG) (220 μL) for 12 h. MII oocytes were collected from the mouse oviduct.

## Human oocytes and morula embryo collection

All donated oocytes and embryos were collected after the consent and approval of the patients based on the guidelines of the ethical committee of the Sixth Affiliated Hospital of Sun Yat-Sen University, Provincial Hospital, Affiliated to Shandong University, First Affiliated Hospital of Nanjing Medical University, First Affiliated Hospital of Wenzhou Medical University, and Shanghai Tenth People's Hospital of Tongji University and in accordance with the regulations and guidelines of the People's Republic of China on the ethical principles of the Human Assisted Reproductive Technology and the Helsinki declaration (IRB approval number: 2017SZZX-013). All donors were informed that their oocytes or embryos would be used for basic research to understand the mechanism of in vitro oocyte maturation and that the donation would not affect their clinical treatment. Patients who underwent assisted reproductive technology (ART) treatment between July 2017 and March 2021 were enrolled in the study. Conventional protocols, including the antagonist and long agonist protocols, were applied for controlled ovarian hyperstimulation (COH). Transvaginal ultrasound-guided oocyte retrieval was performed 36 ± 2 h after the hCG trigger. Cumulus–oocyte complexes (COCs) were exposed to 80 IU/mL of hyaluronidase (SAGE, Cooper Surgical Inc., USA) for 30–45 s and then mechanically denuded using 155-μm stripper pipettes (Research Instruments Ltd., UK). Microscopes (magnification of ×200–×400) were used to assess the integrity and maturity of the denuded oocytes. The vitrification kits were obtained from KITAZATO (KITAZATO Vitrification Kit, Code VT120). Oocytes were rinsed in a droplet of HEPES solution containing 20% serum plasma substitute (SPS) for 1 min; then the droplets of HEPES and solution A were connected to a line for 3 min. Immediately after the droplet of HEPES and a new droplet of solution A were connected to a line for 3 min. Next, the oocytes were transferred into solution B and immersed for 5 min. Finally, they were transferred into a new droplet of solution B, placed onto the cryotip within 90 s, and frozen in liquid nitrogen. For thawing stored oocytes, the cryotip was removed from the liquid nitrogen, and immediately placed in a 1.0 M sucrose thawing solution at 37 °C heating stage for 1 min, and rinsed in serial sucrose solutions for 3 min each (0.75, 0.5, and 0.2 M). Next, they were placed in HEPES solution with 10% SPS for 3 min. Finally, the oocytes were individually transferred into 20 mL of G1 medium (Vitrolife Sweden AB, Goteborg, Sweden).

All human GV and MII oocytes and morula embryos were collected from donors under 35-year-old using a standard protocol to minimize the potential aging effect on oocyte maturation. To minimize the individual heterogeneity resulting from individual donors and handling time, we collected two replicates of 10 oocytes (20 each for GV and MII) from different donors for the general T&T-seq dataset. In OOSP2 functional assays, we collected several pairs of GV oocytes from the same donors. Each time-lapse experiment and single-oocyte T&T-seq were conducted by comparing each pair of oocytes to minimize individual heterogeneity. Three morula embryos were collected from three ICSI patients, who had at least six good-quality embryos, with financial compensation.

## Preparation of functional RiboLace beads

RiboLace beads (RL001, Immagina) were prepared according to the manufacturer's protocol. In brief, 90 μL of beads were added into a 1.5 ml tube. The tube was then placed on a magnet to separate the beads. The beads were washed for 5 min with an equal volume of 90 μL OH-buffer. Then the supernatant was removed and the beads were washed with 900 μL of nuclease-free water. The beads were then washed twice with 90 μL of B-buffer two times. The beads were resuspended in 30 μL of the RiboLace smart probe and incubated for 1 h at room temperature (RT) in a shaker at 1400 rpm. Next, they were blocked with mPEG (3 μL) by agitation at RT for 15 min. Finally, beads were magnetically collected, washed with 500 μL nuclease-free water, 500 μL of W buffer (2 times), resuspended in 100 μL of W buffer, and stored at 4 °C.

## Sample preparation for T&T-seq

For T&T-seq, a single oocyte sample contained only one mouse and one human GV/MII oocyte; a single embryo sample contained the whole morula embryo without further dissection. Oocytes/embryos were transferred to a drop of acid Tyrode's solution (T1788, Sigma-Aldrich) using an oral pipette for 30 s to remove the *zona pellucida*. Oocytes/embryos were then transferred to phosphate-buffered saline (PBS) drop and washed 2 times. The cells were lysed in 10 μL of ice-cold lysis buffer (Single Cell Full Length mRNA-Amplification Kit, N712, Vazyme) for 10 min. Lysates were divided into two parts: 2 μL were used for transcriptome library preparation and thus directly reverse transcribed using a Single Cell Full-Length mRNA-Amplification Kit (N712, Vazyme) following the protocol provided by the manufacturer. The remaining 8 μL were used for translatome library preparation.

293FT cells were detached with TrypLE (12605, Thermo Fisher Scientific) and washed with PBS. Cells were counted using a blood cell counting plate. Cell suspensions were then serially diluted with PBS drops. Finally, single 293FT cells were collected using an oral pipette under stereoscopic guidance.

## Translating mRNAs enrichment for T&T-seq

For mRNA translation assays, 8 μL of lysates were mixed with 8 μL of binding buffer (20 mM Tris–HCl pH 8.0, 150 mM NaCl, 5 mM $MgCl_2$, 1 mM DTT, 100 μg/μL cycloheximide). Functional RiboLace beads (10 μL) were magnetically separated in 1.5 mL tubes and resuspended in 16 μL of lysate. The mixture was incubated for 70 min on a rotator mixer at 10 rpm at 4 °C. The beads were washed twice with 500 μL of W-buffer (RL001, Immagina) on a magnetic stand at 4 °C. To release ribosome-bound mRNAs, 20 μL (1%) sodium dodecyl sulfate (SDS) and 5 μL (0.1 mg) of proteinase K were added to the washed beads, followed by incubation at 37 °C for 30 min. The released RNAs were purified using 1 mL TRIzol (15596026, Invitrogen) in phasemaker tubes (A33248, Invitrogen) and recovered by isopropanol precipitation overnight. The next day, RNA precipitates were dissolved in 1 μL dNTP Mix, 1 μL Oligo (dT) VN Primer, and 3.5 μL of sample buffer, all contained in the Single Cell Full-Length mRNA-Amplification Kit (N712, Vazyme); reverse transcription was performed immediately.

## Library preparation for T&T-seq

Full-length cDNAs for both the transcriptome and translatome were amplified using a single-cell full-length mRNA-amplification kit during 18 PCR cycles. Then, cDNA products were purified using VAHTS DNA Clean Beads (N411, Vazyme). One microliter of the purified cDNA amplification products was assayed using an Agilent 2100 Bioanalyzer and a high-sensitivity DNA chip (Agilent Technologies). The peak of the library was approximately at 2000 bp. The library was constructed using TruePrep® DNA Library Prep Kit V2 for Illumina® (TD502/503,

Vazyme). Purified cDNA amplification products (1 or 5 ng) were used for library fragmentation. Then, library fragments were amplified by PCR during 12 or 15 cycles (5 ng for 12 cycles and 1 ng for 15 cycles). The amplified products were sorted by length using VAHTS® DNA Clean Beads, and 250–450 bp libraries were selected. Library quality was tested using Qubit, quantitative real-time PCR, and Agilent 2100 Bioanalyzer. Libraries were sequenced using the Novaseq 6000 platform. For each sample, we sequenced approximately 20 million reads from both the transcriptome and translatome.

### MiniRibo-seq

The Ribo-seq protocol was modified according to a previous study using $2.5 \times 10^6$ cells or approximately 10 μg of RNA[38]. After optimization, we lowered the input to 50 mouse oocytes. Fifty denuded mouse oocytes were lysed in Ribo-seq lysis buffer (1% Triton X-100, 20 mM Tris–HCl pH 8.0, 150 mM NaCl, 5 mM MgCl₂, 1 mM DTT, and 100 μg/μL cycloheximide). The total DNA and RNA were digested by a mix of 1 U TURBO™ DNase (AM2239, Invitrogen), 0.02 U RNase A, and 0.8 U RNase T1 Cocktail (AM2286, Invitrogen) at 37 °C for 30 min. The digestion was terminated using SUPERase•In™ RNAase inhibitor (AM2696, Invitrogen). RNAs were purified using TRIzol (15596026, Invitrogen) in phasemaker tubes (A33248, Invitrogen), and the resulting RNAs were treated with T4 PNK (M0201S, NEB) to conduct end-repair. Ribosome-protected fragments (RPFs) were separated using 40% RNA PAGE-gel. Resulting 26–40 nt bands were cut and minced using Squisher-Single (H1001, ZYMO) and soaked in recovery buffer (300 mM NaOAc pH 5.5, 1 mM EDTA, 0.25% v/v SDS, 10 mM MgCl₂) overnight. The RPFs were recovered by isopropanol precipitation. Libraries were then constructed using a SMARTer® smRNA-Seq Kit for Illumina (635030, TAKARA). Finally, samples were quantified using the Bioanalyzer High-Sensitivity DNA assay (Agilent Technologies) and sequenced using the Novaseq 6000 platform.

### Dual-luciferase reporter assay

The dual-luciferase reporter assay was performed as previously described[39]. Briefly, the 3′-UTR of human OOSP2 was cloned into the psiCHECK2 vector (Promega, C8021) according to sequences downloaded from the NCBI website. The primers used are shown in Supplementary Data 13. The relative luciferase activity was calculated by normalizing the values of *Renilla luciferase* activity obtained in control samples measured using the Dual-Luciferase Reporter Assay System according to the manufacturer's instructions (Promega, E1910).

### OOSP2 Trim-Away experiment in human oocytes

To test the knockout effect of the OOSP2 protein, we followed the protocol for Trim-Away experiment[40]. OOSP2 antibody (17343-1-AP, Proteintech) and control immunoglobulins (IgG) (FNSA-0106, Finetest) were purified in advance. For constitutive Trim-Away in immature human oocytes, 2 pL of mRNA and 2 pL of antibodies were co-injected. All antibodies were microinjected at a concentration of 200 μg/mL with 600 ng/μL of Trim21 mRNA. Target proteins were depleted for 3 h in the IBMX medium before the oocytes were released. The cells were then subjected to in vitro maturation.

### In vitro maturation experiment of human oocytes

A total of 102 oocytes were collected from 40 donors (some donors donated more than 2 oocytes). Each pair of oocytes from the same donor were divided into two groups, experimental and control. To test the inducing effect of OOSP2, hRec-OOSP2 (Origene) was added to the IVF medium (G-IVF™ PLUS, Vitrolife) at a final concentration of 150 ng/mL. To test the inhibitory effect of the OOSP2 antibody, 1:100 diluted OOSP2 antibody (Abcam) was added to the IVF medium, and the same dilution of IgG (12-370, Millipore) was added to the control IVF medium.

Each oocyte was cultured separately in 25 μL medium covered with mineral oil. Oocytes were incubated inside independent chambers in EmbryoSlide culture dishes (Vitrolife), designed for live imaging. Each pair of oocytes were subjected to live imaging at the same starting point in a humidified atmosphere of 6% CO₂ at 37 °C, in the Embryoscope Time-lapse System (Vitrolife). Phase-contrast images were obtained simultaneously for each pair of oocytes every 15 min for approximately 72 h.

To examine the immediate effect of hRec-OOSP2 and OOSP2 antibody on transcriptomes and translatomes by the T&T-seq procedures described above, GV oocytes were treated with hRec-OOSP2 or OOSP2 antibody for 1.5 h in IVF media at 37 °C, 6% CO₂, in a humidified atmosphere and a regular embryo incubator.

### Quantifications and statistical analyses

Statistical details, including the statistical test, used, the value of replicate, and statistical significances, are stated in the figure legends. GraphPad Prism 6 software was used to perform all statistical analyses using the Student's *t*-test, assuming equal variance and a normal distribution, and *p* values < 0.05 were considered significant in all analyses. All error bars indicate standard deviation (SD). Volcano plots and heatmaps were constructed using R software (R studio version 1.1.456), GraphPad Prism 6, and TBTools (v1.09854). Icons were downloaded from www.biorender.com.

### T&T-seq data processing

The raw reads were trimmed and quality controlled by Trim Galore (v0.6.4) using the following parameters: "trim_galore -q 20–phred33–stringency 3 -j 8–length 50 -e 0.1–paired XX_1.fq.gz XX_2.fq.gz–gzip -o output_path/". Clean reads were mapped using Hisat2 (v2.1.0)[41] against build hg38 of the human genome or mm10 mouse genome, which were downloaded from the Gencode database using the following parameters: "hisat2 -p 8 -x /path/to/genomeindex −1 XX_1.fq.gz −2 XX_2.fq.gz -S XX.sam–summary-file XX.report". The Hg38 and mm10 GTF files were obtained from GenBank. Read counts were calculated by Featurecounts (v1.6.5)[42] using the following parameters: "featureCounts -T 32 -O -t exon -M -a /path/to/gtf–extraAttributes gene_name -o count.txt -p XX.sam". Transcript levels were quantified as TPM using a manual script. For T&T-seq, only protein-coding genes were used for downstream analyses. DEGs were analyzed by DESeq2 (v1.26.0)[43] for genes with TPM > 1. TE was calculated using the TPM of the translatome divided by the TPM of the transcriptome.

### Gene ontology and protein–protein interaction (PPI) analyses

Gene ontology (GO) and PPIs were performed using the Metascape website[44]. Default parameters were used. For GO enrichment, Min Overlap = 3, *p*-value cutoff = 0.01, Min Enrichment = 1.5. For PPI, min network size = 3, max network size = 500, and physical core.

### Principle component analysis (PCA)

For bulk data, PCA was performed on all samples using gene counts of more than 10 in half of the samples using DESeq2. For single-cell data, PCA was performed by Seurat (3.2.3)[45] using the top 2000 most variable genes.

### Pseudotime analysis

Pseudotime analysis was generated by Monocle package (v2.10.1)[27] based on highly variable genes identified by Seurat following the default settings. Genes with TPM > 1 in over half of the samples were used in the pseudotime analysis. Default parameters (dimension reduction method = DDRtree, max_components = 2).

### MiniRibo-seq data processing and analysis

The reads were trimmed with Cutadapt (v1.18)[46] using the following parameters "cutadapt -j 8 -a AAAAAAAAAA XX.fq.gz -o XX_cut.fq.gz -u 3–discard-untrimmed -m 20 -M 40". Only 20–40 nt length reads were

used for downstream analysis. Reads mapped to rRNA were removed by Bowtie2 (v2.3.5.1)[47] using the following parameters "bowtie2 -p 16 -x /path/to/rRNAindex -U XX_cut.fq.gz–un XX_norRNA.fq.gz -S rRNA.align–norc". The remaining reads were mapped to hg38 genome by using STAR (v2.7.1a)[48] using the following parameters "STAR–outFilterType BySJout–runThreadN 8–outFilterMismatchNmax 2–genomeDir /path/to/genomeindex–readFilesIn XX_norRNA.fq.gz–outFileNamePrefix XX–outSAMtype BAM SortedByCoordinate–quantMode TranscriptomeSAM GeneCounts–outFilterMultimapNmax 1–outFilterMatchNmin 16–alignEndsType EndToEnd–outSAMattr IHstart 0". Unique mapped reads were analyzed by RiboCode (v1.2.11)[49] using default parameters.

## RNA 3′-UTR motif analysis

These scripts are available at: https://github.com/lynhsiong/MotifEnrichForUTR. To perform an exhaustive scanning of RNA motif sites in our target gene lists, we first compiled a unified compendium of RNA motifs. Approximately 2957 RNA motifs were curated for 281 RBPs available from diverse RNA motif databases (including RNACOMPETE[50], RBPDB[51], CIS-RBP, and ATtRACT[52]) and multiple CLIP datasets (including CLIPdb[53], Starbase[54], eCLIP[55]). Then, motif scanning was conducted to search for potential RBP-binding sites of the 3′-UTR of all of our target gene lists using FIMO tools of MEME-suite (5.4.1) (parameter: "–verbosity 1–text –norc")[56]. To improve the enrichment signals for post-computational analysis, we filtered the FIMO results to remove the predicted sites of low scores by normalizing motif scores followed by a strict threshold. Finally, using the clean datasets, we created a custom script to calculate the occurrence of specific motifs in our target datasets and our background control datasets to enrich the target motifs in oocyte maturation processes. A single-tail *t*-test was used for *p*-value estimation.

## Reporting summary

Further information on research design is available in the Nature Research Reporting Summary linked to this article.

## Data availability

The sequencing data generated in this study have been deposited at Gene Expression Omnibus (GEO) under accession number GSE197578. Source data has been provided with this paper. Source data are provided with this paper.

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

## Acknowledgements

The authors thank Prof. Wei Xie for many constructive suggestions and discussion of the datasets; Prof. Jie Qiao, Prof. Qingyuan Sun, Prof. William S.B.Yeung, Prof. Enkui Duan for project supervision. This work was supported by the National Key Research and Development Program of China (2017YFC1001601, 2017YFC1001600); the National Natural Science Foundation of China (82071597); Tsinghua-Peking Center for Life Sciences.

## Author contributions

Conceptualization, W.Hu, K.K., H.Z., and X.L.; methodology, W.Hu, K.K., and C.Z.; clinical sample procurements, Y.S., Y.C., T.L., F.C., K.W., L.C., W.N., Y.L., and Y.Z.; data and statistical analysis, W.Hu, T.X., W.Huang, Q.C.Z., and X.F.; resources, K.K., X.L., Z.C., and R.C.; data curation, W.Hu, J.H., S.X., P.S., and L.J.; writing—original draft, W.Hu and K.K.; writing—review and editing, W.Hu, K.K., H.Z., and X.L.; supervision, K.K., H.Z., and X.L.; funding acquisition, K.K., X.L., Z.C., and R.C.

## Competing interests

The authors declare no competing interests.

## Additional information

**Supplementary information** The online version contains

supplementary material available at

