## [Peer Review File · Nature Communications]

Title: Single-cell transcriptome and translome dual-omics reveals potential mechanisms of human oocyte maturationREVIEWER COMMENTS

Reviewer #1 (Remarks to the Author):

This study reports a single-cell multi-omics sequencing method, which could simultaneously detect the transcriptome and translome in single oocytes. Although the transcriptomes of both human and mouse oocytes have been extensively documented, insights into actively translated mRNAs are still in urgent need. Proteomics of oocytes have been achieved in mice, while the amount of starting cells and limitations in sensitivity impede the applications into non-rodent samples. Thus, the T&T-seq reported in this study represents an alternative tool to dig up information on those genes with both active transcription and translation. The maturation of oocytes is a unique process that is very suitable for revealing the advantages of T&T-seq. By comparisons between human and mouse GV-MII oocytes, the authors showed OOSP2 is specifically involved in human oocyte maturation. Overall, the methodology part is interesting and the biological insight of human OOSP2 is a straightforward reflection of T&T-seq. However, the T&T-seq method needs more validations, and technique concerns still exist. The authors present dual-omics at single-cell level instead of translome alone, but do not sufficiently analyze the relationship between these two omics in the same single cell. While the functional test of OOSP2 in human oocytes is appreciated, the data are preliminary in their current form. The followings are specific comments.

-Major comments

1. Cell type universality. The oocyte is unique for its molecular abundances such as RNAs and proteins, so a "single oocyte" could not typically represent any other cell types in testing a single-cell RNA sequencing method. The authors need to examine their single-cell method in additional cell lines such as embryonic stem cells and somatic cell lines.
2. Manual separation of dual omics. The authors claimed that 20% of single-cell lysates had undergone the single-cell SMARTer protocol to build a transcriptome, while the rest 80% would be further used to construct a single-cell translome. This '20%-80%' scheme lacks experimental rationality. The authors should test the "50%-50%" separation of single-cell lysates to examine if this would significantly change the data quality of T&T-seq. More importantly, as the authors emphasized that an optimized single-cell translome-seq strategy was developed in this study, additional control data that the use of whole single-cell lysates to construct only translome is expected to include and calculate the R-square value with the data from miniRibo-seq or Ribotag.
3. Fidelity of the transcriptome in T&T-seq. Because part of, not all of, single-cell lysates were used for the SMARTer scRNA-seq procedure, it needs to compare the data of transcriptome in T&T-seq with standard SMARTer seq.
4. False positive rate of translome in T&T-seq. One of the important issues is to quantify the false positive rate in translome detection by T&T-seq, as nonspecific bindings may occur during the affinity purification of actively translated mRNAs. One suggestion is to incorporate ERCC spike-ins into single-cell lysates and calculate their TPMs in the translome part to infer the false positive rate.
5. True positive rate of translome in T&T-seq. The authors have done the miniRibo-seq, which may be used as controls to calculate the true positive rate of translome in T&T-seq.
6. The relationship between the transcriptome and the translome in single cells. The authors performed single-cell dual-omics but mainly focused on translome alone. The lack of analysis in the relationship between T&T omics omitted the opportunity for quantitative measurements. For example, what is the correlation coefficient between T&T in the same single cell? It is expected that highly abundant genes may actively be translated. What about genes that are low in abundance but high in translation? The authors should dive deeper into these questions and also provide either experimental validation or literature support.

7. Human-mice comparisons. While the authors performed human-mice oocytes analysis, key information was missing. Was Fig. 4a-c conducted using one-to-one orthologous genes? The authors only compared differentially translated genes in oocyte maturation, although this is welcome. The RBP results in Fig. 3 are interesting and important. The authors could also compare human-mice oocytes in a similar analysis performed in Fig. 3 to identify species-specific regulators.

8. OOSP2 experiments in human oocytes. Knockout of three Oosp1-3 in mouse oocytes had minimal effects on oocyte maturation, while the authors showed that OOSP2 alone may be involved in human oocyte maturation. The identification of factors in human oocyte maturation is of biological and clinical significance, but more solid data and sequential experiments are needed to draw conclusions. Good examples and tools are published papers such as 'Clift et al., Nat Commun, 2015', 'Clift et al., Nat Protoc, 2018' and 'Roeles et al., Nat Commun, 2019'. The addition of hRec-OOSP2 to the IVM medium has a minimal effect in promoting oocyte maturation ($P=0.08596$) as shown in Fig. 6b. Incorporation of OOSP2 antibody into IVM medium resulted in discrepancy among different donors and single oocytes. Thus the validation-test results of OOSP2 in human oocytes are preliminary in their current form. Another suggestion is to include functional experiments (knock-down, knock-out, or endogenous protein degradation) in mouse oocytes based on mouse-specific DEGs identified in Fig. 4c, as these experiments could be more feasibly conducted in mouse oocytes, and fit for validating the results of T&T-seq.

9. Supplementary figures. The authors presented five suppl. figures, unfortunately, these suppl. figures were not informative. The authors need to improve both data quality and size to support their methodology and biology claims.

-Minor comments

10. Fig. 1c: should incorporate miniRibo-seq data and add exact values of R.

11. Fig. 1d: do the authors need to give explanations of why dots of MII (or GV) oocytes were separated from each other by different methods? Even MII oocytes from 10 cells in a batch could not cluster with single oocytes by T&T-seq.

12. Fig. 1f: it is better to calculate the overlap of the DEGs up or down among three groups (50-cells miniRibo-seq, 10-cells T&T-seq, and single-cell T&T-seq).

13. Fig. 1g: the GO term results do not reflect a consistent result among the three groups.

14. Fig. 2a: it is better to add Oosp1-3 expression based on proteomics data from mouse oocytes and compare it with that from the T&T-seq.

15. Page 7, line 162-164, 'According to... such as TUBB4B': the sentence is not appropriate. There are positive regulators but also negative ones. Besides, Dnmt1 is not necessary for oocyte maturation in mice.

16. Fig. 7: DEGs between GV oocytes and ab-OOSP2 ones should also be analyzed.

Reviewer #2 (Remarks to the Author):

Hu et al. present a technology called T&T seq, referring to a dual-omics method for profiling both the transcriptome and translome in parallel, using low input material such as the oocytes profiled in this study, but it is significant to note that this technology could be applied to any cell type in that sense. The technology is also significant in that one can delineate T&T features from both the nuclear and

cytoplasmic compartments for downstream functional analysis using data produced with T&Tseq. Moreover, this is clearly necessitated by the need to disambiguate gene expression and transcriptome in whole, while most traditional technologies either fall short on the input requirement and/or need to integrate these data in-silico and/or need for targeted approaches on either the side of gene or protein expression when in parallel. This manuscript focuses on leveraging the dual-omics technology to profile the transcriptome and transcriptomes of maturing human and mouse oocytes from the GV to MII stage and compares those profiles to morula embryos for developmental biology purposes. In all, OOSP2 is identified, for the first time, as a key induction factor for early oocyte maturation and demonstrates the power of this method given the need for comparing gene and protein expression in the case of OOSP2, given the binding predictions and human-specific UTR CPE motifs for human versus the oosp2 mouse analog. Functional validation with OOSP2-blocked (using antibody) experiments versus control successfully shows control over oocyte maturation in developing embryos, albeit with some degree of variability.

Overall, T&Tseq is a useful method that essentially integrates existing RiboLace technology with SMARTerSeq to enable the platform, making it more of a technology advancement rather than a purely novel method, though is robustly integrated and demonstrated to enable novel findings in the OOSP2 example and also notable that competing technologies such as miniRiboSeq and RiboTag require much higher input preventing these types of single to few-cell cohorts to be profiled using existing methods without the need for in-silico data integration. I think the manuscript would require the following revision and authors to address the following questions prior to consideration for publication:

1. In the Figure 2 data and overall results and discussion, the discordance between the transcriptome and transcriptome is highlighted and well taken. However, there is no discussion of the fact that the transcriptome is inherently going to have an expected degree of non-coding potential and this should be brought to light in the discussion especially due to plots like Figure 2b which suggests that one would have a tough time disambiguating features from GV and MII staging which may not be the case once non-coding vs coding predictions are leveraged in context using tools like CPAT to filter for coding potential and/or ORF features as well as Pfam to predict known protein domains to then better filter coding vs noncoding to consider in the transcriptomics side. This is important because inherently the non-coding and coding potential is a feature of transcriptomics (not a fault). If the authors could better address this, it would provide a clearer understanding of the benefits of T&T.
2. Can the authors clarify if the GV and MII oocytes truly have no overlap as shown in the venn diagrams in Figure 2a? If that is just an artifact of doing the GV and MII analyses separately, then it would be beneficial to show the global analysis and overlap between stages as opposed to the two separately as illustrated in 2a, now.
3. Lines 133 to 136 demonstrate the power of T&Tseq, though the authors should comment on expected diversity given the number of single oocytes used in this study vs batches of 10 was limited and given natural oocyte maturation diversity, do they expect some degree of drop out due to under sampling and how to address oocyte or single-cell ensembles versus one by one. While sensitivity is clearly better, the N would need to be much higher for running cell by cell.
4. In the methods section, the authors do not state the NovaSeq per sample read depth that was used which is important for understanding the level of sensitivity on the transcriptomics part of the analysis as related to items #1 and #2 above, it's important to know if the data were limited due to read depth for the DGEA analytics.
5. Why were morula used in this study as an embryonic proxy instead of zygote or an early stage of development? Morula are often known as one of the more diverse stages of preimplantation embryo development given activation of gene expression and splicing happening prior to that stage which would inherently expect the results and separation seen in the pseudotime and overall analysis versus

more relevant earlier stages of development. Can the authors comment on this in more detail in context of lines 304 to 305.

6. Can the authors comment on whether the “embryo” sequencing was truly single cell, meaning the morula sample was disaggregated and T&T was performed on each cell, one by one, or if they performed T&T on the morula as a multi-cellular entity? If the later, this should be more clearly stated in the methods and results/discussion which imply it’s a single cell dataset. This is important due to the different transcriptome and transcriptomics of each cell in the morula when comparing these data.

7. In Figure 2h, I and j, can the authors include more rigorous discussion of the statistical analysis and regression between the 10 oocyte and single oocyte T&T features. While the text suggests they are highly similar between 10 v 1, the variability of single oocyte expression and transcriptome features is quite high when looking at oocyte to oocyte features on a gene-specific local level. Also, in some cases these features seem to be quite different (up vs down regulated in some cases) between oocytes and between 10 v 1 biologic replicates, which should be described in more detail to avoid ambiguity of small N.

8. Figure 3e is fine but would also be interesting to see presented as the TE metric as well as using some motif binding predictions and master regulatory predictions of tools like TOBIAS if/when ATACseq data is available on these oocytes whether publicly available or prospective.

9. Again in Figure 4 a through c, it would be useful to consider or potentially adjust for the non-coding vs coding predictions for the DEG aspect of this analysis.

10. Figure 5 and the associated analysis is quite interesting and well received. It would be interesting to see how OOSP2 and oosp2 are connected through a WGCNA or similar network approach just to better understand association with the other highly secreted and/or highly expressed oocyte maturation factors that are known and less established.

11. Lines 259 through 262 is fine but looking at the data from Figure 6, especially 6e and 6f, the authors should discuss more about the inherent diversity in the oocyte maturation process in control vs Ab-OOSP2 vs hRec-OOSP2. I would expect there to be strong donor dependency and how was this adjusted for given the small sample N.

12. For the pseudotime Monocle analysis in Figure 7b, shouldn’t there be more order to the trajectory from Ab-OOSP2 (blocked) or GV to hRec-OOSP2 to MII to morula as opposed to a lineage split between MII and morula? The clustering looks good and just some clarity on the pseudotime and how it was parameterized to achieve the current figure plot in 7b would be useful.

13. In the methods, the authors should comment on known or expected oocyte collection variability (handling and how this might impact positively or negatively gene expression due to early conditioning and how it was controlled for or considered in the selection and analyses)

14. Overall, the bioinformatics section of the methods is somewhat vague and should be described in more details.

15. A minor suggestion to reword the statement “translated mRNAs and total mRNAs” in the abstract as it is somewhat confusing.

With these edits and considerations adjusted, the manuscript should be more amenable for publication.

Responses to reviewers (NCOMMS-22-05327)

Reviewer #1 (Remarks to the Author):

This study reports a single-cell multi-omics sequencing method, which could simultaneously detect the transcriptome and translome in single oocytes. Although the transcriptomes of both human and mouse oocytes have been extensively documented, insights into actively translated mRNAs are still in urgent need. Proteomics of oocytes have been achieved in mice, while the amount of starting cells and limitations in sensitivity impede the applications into non-rodent samples. Thus, the T&T-seq reported in this study represents an alternative tool to dig up information on those genes with both active transcription and translation. The maturation of oocytes is a unique process that is very suitable for revealing the advantages of T&T-seq. By comparisons between human and mouse GV-MII oocytes, the authors showed OOSP2 is specifically involved in human oocyte maturation. Overall, the methodology part is interesting and the biological insight of human OOSP2 is a straightforward reflection of T&T-seq. However, the T&T-seq method needs more validations, and technique concerns still exist. The authors present dual-omics at single-cell level instead of translome alone, but do not sufficiently analyze the relationship between these two omics in the same single cell. While the functional test of OOSP2 in human oocytes is appreciated, the data are preliminary in their current form. The followings are specific comments.

We thank reviewer#1 for recognizing the value and significance of our study. We have addressed and answers the reviewer's concerns and questions in the following paragraphs:

-Major comments

1. Cell type universality. The oocyte is unique for its molecular abundances such as RNAs and proteins, so a “single oocyte” could not typically represent any other cell types in testing a single-cell RNA sequencing method. The authors need to examine their single-cell method in additional cell lines such as embryonic stem cells and somatic cell lines.

Response: We appreciate Reviewer #1 suggestion to expand the single-cell T&T-seq to other cell types which could be more useful for researchers in other fields. We tested and optimized T&T-seq in 293FT cells and finally succeeded in reducing the cell numbers to single cell. We tested T&T-seq using 1000, 100, 10 and 5 replicates of single 293FT cell. R correlation coefficients of the single-cell samples of transcriptome showed 0.83 to 0.92 and translome showed 0.80 to 0.92 compared to the 1000 cell sample (Response Figure 1). (related to Fig 1c, main text line 99-101 in the revised manuscript)

Response Figure 1. R correlation heatmap of dual-omic for different number of 293FT cells.

2. Manual separation of dual omics. The authors claimed that 20% of single-cell lysates had undergone the single-cell SMARTer protocol to build a transcriptome, while the rest 80% would be further used to construct a single-cell translome. This '20%-80%' scheme lacks experimental rationality. The authors should test the “50%-50%” separation of single-cell lysates to examine if this would significantly change the data quality of T&T-seq. More importantly, as the authors emphasized that an optimized single-cell translome-seq strategy was developed in this study, additional control data that the use of whole single-cell lysates to construct only translome is expected to include and calculate the R-square value with the data from miniRibo-seq or Ribotag.

Response: Following the reviewer’s suggestion, we have tested “0-100%”, “20%-80%”, “50%-50%” “100%-0%” separation scheme of cell lysate. We used mouse GV oocytes to conduct the T&T-seq with 2 replicates for each set of samples. After sequencing, we calculated R correlation coefficients and confirmed that all combinations of lysates are highly correlated in both transcriptome and translomes (all $R > 0.96$, Response Figure 2). (Related to Fig 1b, main text line 95-99 in the revised manuscript)

Response Figure 2. R correlation of translomes in the mouse oocyte lysate with different percentage of lysates.

3. Fidelity of the transcriptome in T&T-seq. Because part of, not all of, single-cell lysates were used for the SMARTer scRNA-seq procedure, it needs to compare the data of transcriptome in T&T-seq with standard SMARTer seq.

Response: Following reviewer# 1 suggestion, we have also validated that different percentages of oocyte lysates also yielded highly reproducible transcriptomes. (Response Figure 3). (Related to Fig 1b, main text line 95-99 in the revised manuscript)

Response Figure 3. R correlation of transcriptomes in the mouse oocyte lysate with different percentage of lysates.

4. False positive rate of translome in T&T-seq. One of the important issues is to quantify the false positive rate in translome detection by T&T-seq, as nonspecific bindings may occur during the affinity purification of actively translated mRNAs. One suggestion is to incorporate ERCC spike-ins into single-cell lysates and calculate their TPMs in the translome part to infer the false positive rate.

Response: To address the potential false positive reads of translome, we have incorporated ERCCs spike-ins into oocyte lysates and conducted the T&T-seq. The average ERCCs TPM in transcriptome is 10.99, while the average ERCCs TPM in translome is 1.83, so the false positive rate is about 0.167 which is lower than RiboTag (0.3)¹ (Response Figure 4) and suitable for performing downstream analysis. (Related to Supplementary Fig 1a, main text line 101-104 in the revised manuscript)

Response Figure 4. Estimating false positive rate using ERCC spike in.

5. True positive rate of translome in T&T-seq. The authors have done the miniRibo-seq, which may be used as controls to calculate the true positive rate of translome in T&T-seq.

Response: We appreciate Reviewer#1's suggestion, but the principle of miniRibo-seq and T&T-seq are quite different so it might not be suitable to use miniRibo-seq as controls to calculate the true positive rate of translome in T&T-seq. RNase was used to digest mRNAs in miniRibo-seq, and the ribosome protected fragments (RPFs) are gel purified to construct the library. The RPFs are usually 30nts which might be mapped to multiple sites in the reference genome and filtered away during analysis, so the sequencing depth is much higher for miniRibo-seq to achieve similar gene expression coverage. In addition, it is more difficult to detect the less actively translated genes by Ribo-seq due to the loss during RNase treatment and mapping filtering step.

On the other hand, T&T-seq utilizes puromycin analog binding to the A-site of the ribosomes, which captures the full-length translating mRNAs, so the less actively translated genes are more likely to be captured and mapped. Considering that Ribo-tag utilize the similar principle of affinity purification though the ribosome are tagged with epitopes, we compared the similarity of translomes for these two independent methods (Ribo-tag datasets were obtained from independent study by Conti's group) and calculated the R correlation coefficient between T&T-seq and Ribo-tag (Response Figure 5.). The T&T-seq and Ribo-tag datasets are produced from different oocytes number and preparations in different laboratories, so R correlation of (0.77-0.82) between the two set of samples showed that the T&T-seq translome are well correlated with the reported translated genes in mouse oocytes (Related to Supplementary Fig 1b, main text line 104-106 in the revised manuscript).

Response Figure 5. Heatmap of R correlation coefficient between T&T-seq and Ribo-tag.

6. The relationship between the transcriptome and the translome in single cells. The authors performed single-cell dual-omics but mainly focused on translome alone. The lack of analysis in the relationship between T&T omics omitted the opportunity for quantitative measurements. For example, what is the correlation coefficient between T&T in the same single cell? It is expected that highly abundant genes may actively be translated. What about genes that are low in abundance but high in translation? The authors should dive deeper into these questions and also provide either experimental validation or literature support.

Response: We appreciate the reviewer's comment on the relationship of transcriptome and

translatomes. This is actually the strength of our methodology. We analyzed the relationship of transcriptome and translatomes starting from Figure 2b to 2j and focusing more on the highly translated genes which are more likely to actively participate in oocyte maturation in the original manuscript. We used T&T-seq of 10 oocytes to analyze the relationship between transcription and translation because the variation of gene expression is lower than the single-oocyte datasets. Nevertheless, we have calculated the R correlation of the transcription and translation in the single-oocyte samples according to the reviewer's suggestion. (Response Figure 6a). The analysis shows that the correlation coefficient between T&T in each GV oocyte are higher than that in the MII oocytes. Moreover, transcriptome of GV oocyte have high correlation with the transcriptome of MII oocytes but much lower correlation with the translatomes of MII oocytes. Taken together, these correlations support the notion that transcriptomes maintain the same profiles from GV to MII oocytes due to genome-wide inactivation of transcriptional programs, while translatomes changes significantly during the same period in the same oocytes.

We have divided the genes based on their T&T and GV to MII stages. 606 genes (class I) show low fold change in transcription ($-1 < \log_2FC < 1$) and high fold change in translatome ($\log_2FC > 1$). Those genes are considered to be increased only in the translatome during oocyte maturation. Gene expression heatmap also show that some of those genes (select by GO terms) maintained similar transcriptional level but became highly translated in MII (Response Figure 6c). After calculating the translation efficiency ($TPM_{translatome}/TPM_{transcriptome}$) during oocytes maturation (Response Figure 6d), we also found that in genes translation efficiency are significant higher in MII than GV stages.

To further investigate the transcriptional abundance of the highly translated genes, we plotted the expression of T&T for the actively translated genes in GV and MII separately (Response Figure 6e,f). If the low abundance at transcriptional level is set at $1 < TPM < 10$, then there are about 71 and 279 low-abundance transcripts in GV and MII respectively. (Related to Fig 2c,e,h, Supplementary Fig 4a-c, 5b, main text line 151-156, 171-175 in the revised manuscript)

Response Figure 6: transcriptome and translome conjoint analysis. a. Heat map of R correlation coefficient between T&T of single human oocyte. b. Scatter plot of transcriptome and translome conjoint analysis between GV and MII oocyte. c. gene expression heatmap of class I genes. GO terms are shown on the left. d. TE volcano plot shows the Translation efficiency during oocyte maturation. e-f. Expression of transcriptome and translome in high TE genes. e, GV oocytes; f, MII oocytes.

7. Human-mice comparisons. While the authors performed human-mice oocytes analysis, key information was missing. Was Fig. 4a-c conducted using one-to-one orthologous genes? The authors only compared differentially translated genes in oocyte maturation, although this is welcome. The RBP results in Fig. 3 are interesting and important. The authors could also compare human-mice oocytes in a similar analysis performed in Fig. 3 to identify species-specific regulators.

Response: Yes, our human-mice comparisons are conducted using one-to-one orthologous genes according to the Ensembl database. (<https://www.ensembl.info/2009/01/21/how-to-get-all-the-orthologous-genes-between-two-species/>). In addition, we also analyzed mouse RBPs during oocytes maturation using TE>1 genes. we identified 10 RBPs in mouse MII oocytes ($\log_2FC>1$, $p<10^{-5}$, Response Figure 7). (Related to Supplementary Fig 8, main text line 248-250 in the revised manuscript)

RBP name	Motif	Log ₂ FC	p value
SRSF2		1.8365	3.98E-12
nElavl		1.65577	6.37E-09
SRSF4		1.60991	4.17E-07
CELF4		1.48712	7.37E-07
RBFox2		1.29572	1.08E-06

Response Figure 7. RBP motifs in mouse MII high TE genes. $\log_2FC>1$, $p<10^{-5}$

8. OOSP2 experiments in human oocytes. Knockout of three Oosp1-3 in mouse oocytes had minimal effects on oocyte maturation, while the authors showed that OOSP2 alone may be involved in human oocyte maturation. The identification of factors in human oocyte maturation is of biological and clinical significance, but more solid data and sequential experiments are needed to draw conclusions. Good examples and tools are published papers such as 'Clift et al., Nat Commun, 2015', 'Clift et al., Nat Protoc, 2018' and 'Roeles et al., Nat Commun, 2019'. The addition of hRec-OOSP2 to the IVM medium has a minimal effect in promoting oocyte maturation ($P=0.08596$) as shown in Fig. 6b. Incorporation of OOSP2 antibody into IVM medium resulted in discrepancy among different donors and single oocytes. Thus the validation-test results of OOSP2 in human oocytes are preliminary in their current form. Another suggestion is to include functional experiments (knock-down, knock-out, or endogenous protein degradation) in mouse oocytes based on mouse-specific DEGs identified in Fig. 4c, as these experiments could be more feasibly conducted in mouse oocytes, and fit for validating the results of T&T-seq.

Response: As shown in Figure 5a, the expression of Oosp2 is the lowest among the three Oosp family in mice, so it would be difficult to show the functional requirement of OOSP2 in mouse oocyte by knockout or knockdown experiments. In contrast, human OOSP2 is the highest expressed member and showing an increasing trend during oocyte maturation. Thus, it would be

more feasible to show additional evidence of the promoting effect on human maturations. We attempted the Trim-away methodology suggested by the reviewer to test if knockdown of OOSP2 may reduce the maturation rate of human oocytes. We found that the protocol is quite challenging for human oocyte because injection of concentrated antibodies into the donated GV oocytes often resulted in degraded oocytes. After some optimizations, we were able to completed 14 sets (each set contain two GV oocytes from the same patients) of Trim-away experiments. According to the published Trim-away protocol², the targeted protein of knockdown requires some reaction time after injection of TRIM21 protein and to ensure enough the degradation of targeted proteins. Hence, we have to add a step of culturing the GV oocytes in 3-isobutyl-1-methylxanthine (IBMX) medium for 3 hours to ensure the OOSP2 protein would be fully degraded (This step was not applied in the previous experiments of maturations). IBMX medium has been used to promote synchronization of nuclear-cytoplasmic maturation which ultimately lead to the increase of human oocyte maturation^{3,4}, so the overall oocyte maturation rate is higher than previous experiments in Figure 6a-d. Finally, 30% of the Trim-away group are not matured after 48 hours compared with 100% maturation rate in the control group using IgG for injection (Response Figure 8a). In addition, we also conducted single-oocyte T&T-seq for the OOSP2 Trim-away oocytes and control oocytes (2 sets each). Genes required for oocyte maturation such as ZP3, GDF9 and OOEP are significantly down-regulated after OOSP2 Trim-away (Response Figure 8b). We also conducted GO enrichment analysis using down-regulated genes after OOSP2 Trim-away (Response Figure 8c). Terms such as “organelle localization”, “female gamete generation”, “chromosome segregation” which are involved in oocyte maturation are enriched in those down-regulated genes. Moreover, we also confirmed the OOSP2 are expressed in human oocytes by immunostaining and the protein is predicted to interact with BMP15, H1FOO by protein network analysis (related to Supplementary Figure 10,11, main text line 293-301 in the revised manuscript). In summary, we have provided several evidences (related to Fig 6,7) in the original manuscript and new evidences (Trim-away, immunostaining, network analysis), all indicated that OOSP2 functions to induce human oocyte maturation.

Response Figure 8: OOSP2 Trim-away experiment. a. Maturation rate in OOSP2 trim-away group and control group. oocytes are collected in pair to reduce individual differences. b. volcano plot shows DEGs after OOSP2 trim-away. c. GO enrichment of the genes down-regulated after OOSP2 Trim-away.

9. Supplementary figures. The authors presented five suppl. figures, unfortunately, these suppl. figures were not informative. The authors need to improve both data quality and size to support their methodology and biology claims.

Response: We have provided more information and datasets so that there are total 12 suppl. figures and total 13 suppl. tables to support our main results and claims.

-Minor comments

10. Fig. 1c: should incorporate miniRibo-seq data and add exact values of R.

Response: We have shown the R values is all R-heatmap (Related to Supplementary Fig.1b; Supplementary Fig. 4b).

11. Fig. 1d: do the authors need to give explanations of why dots of MII (or GV) oocytes were separated from each other by different methods? Even MII oocytes from 10 cells in a batch could not cluster with single oocytes by T&T-seq.

Response: The main purpose of Fig. 1d is to test if the translomes of mouse GV and MII oocytes can be distinguished by the three methods, and if the GV or MII oocytes are clustered closer regardless of the methods used. The PCA analysis show that all GV oocytes clustered to one side and all MII oocytes clustered in the opposite sites. However, the distances among the dots of three methods varies, more significantly in MII oocytes. These are due to several reasons. First, the principle and procedures of the 3 methods are different. Ribo-seq involves the RNase

treatment to digest mRNAs and only ribosome protected fragment are purified for library construction, so the sequenced length is about 30-nts. While the Ribotag and T&T-seq purified full-length mRNAs for library construction, so they cover more gene with lower sequencing depth. In addition, Ribotag expressed affinity tags on the large subunit of ribosomes but T&T-seq uses puromycin analog binding to the A site of the actively translating ribosomes, so Ribotag and T&T-seq are more similar in principle but still may capture some different genes. Therefore, the translomes from three methods are expected to have some distances even though the oocytes are at the same stages. Second, the translomes of MII are inherently more heterogenous than GV oocytes. This may be due to the translational regulation is more active in MII oocyte and the oocytes are not synchronized as during the oocyte maturation process. These are supported by our single-oocyte T&T-seq dataset in which transcriptomes and translomes of GV oocyte are highly correlated but the transcriptomes of MII oocytes are more correlated than the translomes. Therefore, the three replicates of 10 oocyte MII dots cluster closer than the single-oocyte replicates because the 10 oocyte pooling reduce the individual oocyte variation.

12. Fig. 1f: it is better to calculate the overlap of the DEGs up or down among three groups (50-cells miniRibo-seq, 10-cells T&T-seq, and single-cell T&T-seq).

Response: Following reviewer's suggestion, overlap of the DEGs are analyzed and presented as in Response Figure 9. (Related to Supplementary Fig.2b).

Response Figure 9. Venn plot show the overlap among three groups.

13. Fig. 1g: the GO term results do not reflect a consistent result among the three groups.

Response: Although the gene count and p-value might vary among the three groups, the GO term are highly similar, including the enriched GO terms “ribosome biogenesis”, “mitochondrial protein complex”, “rRNA metabolic process” in GV oocytes and “DNA recombination”, “chromosome region”, “regulation of cell cycle process” in MII oocytes.

14. Fig. 2a: it is better to add Oosp1-3 expression based on proteomics data from mouse oocytes and compare it with that from the T&T-seq.

Response: Thanks reviewer's suggestion. We found a study of mouse proteomics by Li et al, 2020⁵. Consistent with our T&T-seq data, they reported that Oosp1 and Oosp3 are upregulated during oocyte maturation. (Oosp1: MII/GV FC= 62.90049576, q value=0.001623188; Oosp3: MII/GV

FC= 47.2065249, q value=0.012457243), but Oosp2 were not detected in their study.

15. Page 7, line 162-164, 'According to... such as TUBB4B': the sentence is not appropriate. There are positive regulators but also negative ones. Besides, Dnmt1 is not necessary for oocyte maturation in mice.

Response: We have modified the sentence to “According to the analysis, genes such as CPEB4, WEE2, DAZL, and DNMT1 which had TE>1 in MII while TUBB4B had TE>1 in GV”. As for “Dnmt1”, we agree that Dnmt1 is not necessary for oocyte maturation in mice. Noticeably, our T&T-seq data shows that Dnmt1 have low translational expression during mouse oocyte maturation, but shows a higher translational trend in human oocyte. This might reflect the different translational expression of specific genes between mouse and human (Response Figure 10). (Related to Supplementary Fig.7b, main text line 181-182 in the revised manuscript).

Response Figure 10. The expression of DNMT1 in the translome in human and mouse oocytes. m: mouse; h: human.

16. Fig. 7: DEGs between GV oocytes and ab-OOSP2 ones should also be analyzed.

Response: We have analyzed DEGs between GV and ab-OOSP2 treated GV. There are xx genes downregulated, including RBM44, TTL, OR1E1 and etc. (Response Figure 11).

Response Figure 11. Volcano plot of DEGs between GV oocytes and ab-OOSP2 ones.

Reviewer #2 (Remarks to the Author):

Hu et al. present a technology called T&T seq, referring to a dual-omics method for profiling both the transcriptome and translome in parallel, using low input material such as the oocytes profiled in this study, but it is significant to note that this technology could be applied to any cell type in that sense. The technology is also significant in that one can delineate T&T features from both the nuclear and cytoplasmic compartments for downstream functional analysis using data produced with T&Tseq. Moreover, this is clearly necessitated by the need to disambiguate gene expression and translome in whole, while most traditional technologies either fall short on the input requirement and/or need to integrate these data in-silico and/or need for targeted approaches on either the side of gene or protein expression when in parallel. This manuscript focuses on leveraging the dual-omics technology to profile the translome and transcriptomes of maturing human and mouse oocytes from the GV to MII stage and compares those profiles to morula embryos for developmental biology purposes. In all, OOSP2 is identified, for the first time, as a key induction factor for early oocyte maturation and demonstrates the power of this method given the need for comparing gene and protein expression in the case of OOSP2, given the binding predictions and human-specific UTR CPE motifs for human versus the oosp2 mouse analog. Functional validation with OOSP2-blocked (using antibody) experiments versus control successfully shows control over oocyte maturation in developing embryos, albeit with some degree of variability.

Overall, T&Tseq is a useful method that essentially integrates existing RiboLace technology with SMARTerSeq to enable the platform, making it more of a technology advancement rather than a purely novel method, though is robustly integrated and demonstrated to enable novel findings in the OOSP2 example and also notable that competing technologies such as miniRiboSeq and RiboTag require much higher input preventing these types of single to few-cell cohorts to be profiled using existing methods without the need for in-silico data integration. I think the manuscript would require the following revision and authors to address the following questions prior to consideration for publication:

We thank reviewer#2 for recognizing the value and significance of our study. We have addressed and answers the reviewer's concerns and question in the following paragraph:

1. In the Figure 2 data and overall results and discussion, the discordance between the transcriptome and translome is highlighted and well taken. However, there is no discussion of the fact that the transcriptome is inherently going to have an expected degree of non-coding potential and this should be brought to light in the discussion especially due to plots like Figure 2b which suggests that one would have a tough time disambiguating features from GV and MII staging which may not be the case once non-coding vs coding predictions are leveraged in context using tools like CPAT to filter for coding potential and/or ORF features as well as Pfam to predict known protein domains to then better filter coding vs noncoding to consider in the transcriptomics side. This is important because inherently the non-coding and coding potential is a feature of transcriptomics (not a fault). If the authors could better address this, it would provide a clearer understanding of the benefits of T&T.

Response: We only selected protein-coding genes according to the annotations of Ensembl in Figure 2 analysis and overall results and discussion and we have clarified this point in our modified manuscript. Following reviewer's suggestion, we selected the noncoding RNAs especially lncRNAs to calculate coding potential using CPAT. We identified 100 and 36 lncRNAs which have high coding potential and a high translation efficiency (TPMtranslatome/TPMtranscriptome) in GV and

MII oocytes respectively (Response table 1). The combined T&T-seq and CPAT analysis shows that these lncRNAs might be translated during oocyte maturation.

2. Can the authors clarify if the GV and MII oocytes truly have no overlap as shown in the venn diagrams in Figure 2a? If that is just an artifact of doing the GV and MII analyses separately, then it would be beneficial to show the global analysis and overlap between stages as opposed to the two separately as illustrated in 2a, now.

Response: The Venn diagrams in Figure 2a showed the overlapped translated genes and proteins identified in independent proteomic study of GV and MII oocytes separately. Following reviewer's suggestion, we tried to analyze the overlap between transcriptome and proteomic dataset together (Response Fig 12a). However, this figure looks complicated and didn't visually show the proportion of overlapped gene compared with the Venn diagrams. Therefore, we keep the Venn figure, but add rectangular frames to both Venn diagrams of GV and MII oocytes to separate the two groups (Response Fig 12b). (Related to Fig.2a, main text line 139-146 in the revised manuscript).

Response Figure 12 Venn plots shows the overlap between transcriptome and proteome in GV and MII oocytes. a. the global analysis and overlap between stages. b. separated analysis of two stages.

3. Lines 133 to 136 demonstrate the power of T&T-seq, though the authors should comment on expected diversity given the number of single oocytes used in this study vs bathes of 10 was limited and given natural oocyte maturation diversity, do they expect some degree of drop out due to under sampling and how to address oocyte or single-cell ensembles versus one by one. While sensitivity is clearly better, the N would need to be much higher for running cell by cell.

Response: We thank the reviewer's for pointing out the diversity of gene expressions during oocyte maturation. We agree that the diversity or variation of transcriptomes exist during oocyte maturation due to natural oocyte maturation diversity. These are supported by our analysis of single-oocytes showing that the correlation coefficient between transcriptome and transcriptome in each GV oocyte are higher than that in the MII oocytes (Response Figure 13). Moreover, the transcriptomes among the MII oocytes are highly correlated but the transcriptomes of the same oocytes are much less correlated. This shows that the transcriptomes are inherently more diversified only in MII in which transcriptomes remain the same and highly correlated to each oocyte.

(Related to Supplementary Fig. 4b, main text line 151-156 in the revised manuscript).

Response Figure 13. Heatmaps of R correlation between GV and MII oocytes in transcriptome and translome.

4. In the methods section, the authors do not state the NovaSeq per sample read depth that was used which is important for understanding the level of sensitivity on the transcriptomics part of the analysis as related to items #1 and #2 above, it's important to know if the data were limited due to read depth for the DGEA analytics.

Response: We added that information to “Method” section in our manuscript. We have sequenced about 20M reads per sample. Compared with Ribo-seq, our T&T-seq have a much higher mapping rate (Response Fig. 14) so that we can only need to sequence 20M reads which is about 1/10 of Ribo-seq. Therefore, the sequencing cost of T&T-seq is much lower compared to Ribo-seq. (Related to Supplementary Fig. 1d, main text line 109-111, line 514 in the revised manuscript).

Response Fig. 14 Mapping rates of miniRibo-seq and T&T-seq.

5. Why were morula used in this study as an embryonic proxy instead of zygote or an early stage of development? Morula are often known as one of the more diverse stages of preimplantation embryo development given activation of gene expression and splicing happening prior to that stage which would inherently expect the results and separation seen in the pseudotime and overall analysis versus more relevant earlier stages of development. Can the authors comment on this in more detail

in context of lines 304 to 305.

Response: We agree with reviewer's opinion that zygote or early stage human embryo are less diversified in gene expressions than morula stage, but restricted by the clinical sample procurement protocol, we are only allowed to obtain consented morula stage embryos for this project. In this study, we focus on the oocyte maturation stage, so we mainly focus on collecting the GV and MII oocytes. In the future, we may extend T&T study to embryos in early stages.

6. Can the authors comment on whether the "embryo" sequencing was truly single cell, meaning the morula sample was disaggregated and T&T was performed on each cell, one by one, or if they performed T&T on the morula as a multi-cellular entity? If the later, this should be more clearly stated in the methods and results/discussion which imply it's a single cell dataset. This is important due to the different translome and transcriptomics of each cell in the morula when comparing these data.

Response: We apologize if it is not clear in our last version of manuscript. As we were using the morula stage sample more as the references of embryo-stage profiles, we performed T&T-seq of the morula embryo as a multi-cellular entity. We have clarified this information in our revised manuscript. (main text line 486-488)

7. In Figure 2h, I and j, can the authors include more rigorous discussion of the statistical analysis and regression between the 10 oocyte and single oocyte T&T features. While the text suggests they are highly similar between 10 v 1, the variability of single oocyte expression and translome features is quite high when looking at oocyte to oocyte features on a gene-specific local level. Also, in some cases these features seem to be quite different (up vs down regulated in some cases) between oocytes and between 10 v 1 biologic replicates, which should be described in more detail to avoid ambiguity of small N.

Response: In Figure 2h, i and j, the DEGs are identified using TPM>1 genes from 10 oocytes data, the parameters are p-value<0.05, log2FC>1 or <-1. After this analysis, DEGs were displayed both 10 and single oocytes data. Hence, the single-oocyte datasets might show more variability compared to the 10 oocyte datasets on a gene-specific level. Nevertheless, the global trend is consistent between 10 and single oocyte. These are supported by the evidences that class IV genes are downregulated in both transcription and translation, while in class I, genes are only up regulated in the translome. As we described in Response Fig. 13, translomes of T&T-seq revealed that the inherent diversity was higher in the MII than GV in when we conducted single-oocyte T&T. This was added in our result and discussion in our revised manuscript. (Related to Supplementary Fig. 4b, main text line 151-156).

8. Figure 3e is fine but would also be interesting to see presented as the TE metric as well as using some motif binding predictions and master regulatory predictions of tools like TOBIAS if/when ATACseq data is available on these oocytes whether publicly available or prospective.

Response: In Fig. 3e, we presented transcriptome and translome of the genes containing all 5 RBP motifs at their 3'UTR and show high TE in MII stage. Hence, we are analyzing the potential translational regulation at their 3'UTR. The transcriptomes of these genes remain unchanged from GV to MII but their translomes show significant increase, so their gene expressions are not likely to be regulated by transcription factor. TOBIAS is a bioinformatic tool to analyze the chromatin

accessibility data like ATAC-seq, so it may not be suitable for the translational regulation analysis during oocyte maturation, especially MII oocyte. Nevertheless, we attempted TOBIAS analysis on the reported ATAC-seq data⁶ to look for potential transcription factor in GV oocytes and found xx TFs (Response Fig 15).

Response Fig 15. a. Transcription factors predicted by published GV oocyte ATAC-seq data.

9. Again in Figure 4 a through c, it would be useful to consider or potentially adjust for the non-coding vs coding predictions for the DEG aspect of this analysis.

Response: We only selected protein-coding genes according to the annotations of Ensembl in Figure 4a-c analysis and we have clarified this point in our modified manuscript. Following reviewer's suggestion, we use CPAT to calculate the coding potential of lncRNAs in both human and mouse GV and MII oocytes (Response table 1). We also analyzed the overlap in those high CP lncRNAs between human and mouse. However, there was no overlap lncRNA with high CP between the two species.

10. Figure 5 and the associated analysis is quite interesting and well received. It would be interesting to see how OOSP2 and oosp2 are connected through a WGCNA or similar network approach just to better understand association with the other highly secreted and/or highly expressed oocyte maturation factors that are known and less established.

Response: We have analyzed the protein-protein interaction of human and mouse OOSP2 protein in STRING database (Response Figure 16). The result shows that human OOSP2 protein are correlated to H1FOO, BMP15 which are essential during oocyte maturation. Interestingly, mouse Oosp2 protein network analysis resulted in a different network, showing correlation to Obox5, Oosp1,3 and Zp3 proteins and other proteins. (Related to Supplementary Fig. 9, main text line 269-273 in the revised manuscript).

Response Figure 16 Protein-protein interaction of OOSP2 protein in human and mouse. a. human b. mouse.

11. Lines 259 through 262 is fine but looking at the data from Figure 6, especially 6e and 6f, the authors should discuss more about the inherent diversity in the oocyte maturation process in control vs Ab-OOSP2 vs hRec-OOSP2. I would expect there to be strong donor dependency and how was this adjusted for given the small sample N.

Response: We have considered the inherent diversity and heterogeneity among different donors so that the comparison of control vs Ab-OOSP2 or control-hRecOOSP2 were conducted among each set of two GV oocytes from the same donors. We have stated this normalization in the main text (Line 279-280). This process took us a longer collection time but can reduce the variation. Hence, although the inherent diversity was observed among different donors, the overall trends of maturation are consistent; that is, adding hRecOOSP2 promote the maturation rate and shorten the maturation time whereas adding Ab-OOSP2 decrease maturation rate and increase maturation time. (Related to Figure 6e and f)

12. For the pseudotime Monocle analysis in Figure 7b, shouldn't there be more order to the trajectory from Ab-OOSP2 (blocked) or GV to hRec-OOSP2 to MII to morula as opposed to a lineage split between MII and morula? The clustering looks good and just some clarity on the pseudotime and how it was parameterized to achieve the current figure plot in 7b would be useful.

Response: We used the default parameters of the pseudotime analysis, which is the dimension reduce method= DDRtree, max_components = 2, TPM>1 in over half of the samples. We have added this information to the method section (Line 592-594). This parameter resulted in a split of the morula embryo from the MII and early-stage oocytes. We think that the split might be caused by the developmental stage of the fertilized embryo being significantly different from GV to MII oocytes. Human GV oocytes were incubated with hRec-OOSP2, with ab-OOSP2, or without any additive as control for 1.5 h and subjected to T&T analysis. At the 1.5h incubation time, all oocytes remain at GV stage judging from their GV morphological appearance, so they remain as early stage oocyte relative to MII and the morula embryos. We added a clarification to the text in the revised manuscript. (main text line 308)

Response Fig 17 Experiment design for IVM of human oocytes.

13. In the methods, the authors should comment on known or expected oocyte collection variability (handling and how this might impact positively or negatively gene expression due to early conditioning and how it was controlled for or considered in the selection and analyses)

Response: We collected human GV and MII oocytes from donors under 35 years old by the standard protocol described in details in the method to minimize the potential aging effect on oocyte maturation. To minimize the individual heterogeneity resulted from individual donors and handling time, we collected two replicates of 10 oocytes (total 20 each for GV and MII) from different donors for T&T-seq. These are the main T&T profiles we used for general conclusions. As for the OOSP2 functional assays, we collected many pairs of GV oocyte from the same donors. Each timelapse experiment and single-oocyte T&T-seq was conducted by comparing each pair of oocytes, so the individual heterogeneity would be minimized. (main text line 468-474)

14. Overall, the bioinformatics section of the methods is somewhat vague and should be described in more details.

Response: We have added more details in the bioinformatics section of the methods in the revised manuscript.

15. A minor suggestion to reword the statement “translated mRNAs and total mRNAs” in the abstract as it is somewhat confusing. With these edits and considerations adjusted, the manuscript should be more amenable for publication.

Response: Thanks for reviewer’s suggestion. We have changed the sentence into “we developed a dual-omic methodology which includes both transcriptomes and translomes sequencing (T&T-seq) for single-oocyte and single-cell samples.”

Reference

- 1 Sanz, E., Bean, J. C., Carey, D. P., Quintana, A. & McKnight, G. S. RiboTag: Ribosomal Tagging Strategy to Analyze Cell-Type-Specific mRNA Expression In Vivo. *Curr Protoc Neurosci* **88**, e77, doi:10.1002/cpns.77 (2019).
- 2 Clift, D., So, C., McEwan, W. A., Lames, L. C. & Schuh, M. Acute and rapid degradation of endogenous proteins by Trim-Away. *Nat Protoc* **13**, 2149-2175, doi:10.1038/s41596-018-0028-3 (2018).
- 3 Somfai, T. *et al.* Meiotic arrest maintained by cAMP during the initiation of maturation

- enhances meiotic potential and developmental competence and reduces polyspermy of IVM/IVF porcine oocytes. *Zygote* **11**, 199-206, doi:10.1017/s0967199403002247 (2003).
- 4 Huang, W., Nagano, M., Kang, S. S., Yanagawa, Y. & Takahashi, Y. Prematurational culture with 3-isobutyl-1-methylxanthine synchronizes meiotic progression of the germinal vesicle stage and improves nuclear maturation and embryonic development in in vitro-grown bovine oocytes. *J Reprod Dev* **60**, 9-13, doi:10.1262/jrd.2013-082 (2014).
- 5 Li, L. *et al.* Characterization of Metabolic Patterns in Mouse Oocytes during Meiotic Maturation. *Molecular Cell* **80**, 525-+ (2020).
- 6 Wu, J. *et al.* Chromatin analysis in human early development reveals epigenetic transition during ZGA. *Nature* **557**, 256-260, doi:10.1038/s41586-018-0080-8 (2018).

REVIEWERS' COMMENTS

Reviewer #1 (Remarks to the Author):

For the most part, the authors have addressed my main concerns, especially with respect to techniques. Although technique variations could not be ignored, as shown by the data from 293T single cells. The authors have provided alternative tool and potentially useful resources for analyzing transcription and translation during oocyte maturation. I do have further suggestions for this work.

1. The biological insights and experimental validations are the main weakness of this study. The authors included the Trim-away method, while only 30% (n=3) of the oocytes were abnormal in the extrusion of PB. So, the conclusion that OOSP2 induces human oocyte maturation is over-claimed and should be toned down. Additionally, the title needs to be revised, for example: "Single-cell transcriptome & translome dual-omics analysis of mouse and human oocytes". This would not significantly reduce the novelty of the methodology part.
2. As this is a resource article, the authors need to upload all the sequenced data of this study to their GEO accessions GSE197578, and provide the reviewer link for both referees.
3. A recently published study described a Ribo-lite method for profiling the translational landscapes of mouse oocytes and preimplantation embryos (<https://doi.org/10.1038/s41556-022-00928-6>). May the authors add the Ribo-lite method to their Table 1 and made some comments in the discussion section.

Reviewer #2 (Remarks to the Author):

Hu et al. present a technology called T&T seq, referring to a dual-omics method for profiling both the transcriptome and translome in parallel, using low input material such as the oocytes profiled in this study, but it is significant to note that this technology could be applied to any cell type in that sense. The technology is also significant in that one can delineate T&T features from both the nuclear and cytoplasmic compartments for downstream functional analysis using data produced with T&Tseq. Moreover, this is clearly necessitated by the need to disambiguate gene expression and translome in whole, while most traditional technologies either fall short on the input requirement and/or need to integrate these data in-silico and/or need for targeted approaches on either the side of gene or protein expression when in parallel. This manuscript focuses on leveraging the dual-omics technology to profile the translome and transcriptomes of maturing human and mouse oocytes from the GV to MII stage and compares those profiles to morula embryos for developmental biology purposes. In all, OOSP2 is identified, for the first time, as a key induction factor for early oocyte maturation and demonstrates the power of this method given the need for comparing gene and protein expression in the case of OOSP2, given the binding predictions and human-specific UTR CPE motifs for human versus the oosp2 mouse analog. Functional validation with OOSP2-blocked (using antibody) experiments versus control successfully shows control over oocyte maturation in developing embryos, albeit with some degree of variability.

Overall, T&Tseq is a useful method that essentially integrates existing RiboLace technology with SMARTerSeq to enable the platform, making it more of a technology advancement rather than a purely novel method, though is robustly integrated and demonstrated to enable novel findings in the OOSP2 example and also notable that competing technologies such as miniRiboSeq and RiboTag require much higher input preventing these types of single to few-cell cohorts to be profiled using existing methods without the need for in-silico data integration.

I appreciate the thorough and thoughtful responses and adjustments as suggested. The Authors did a very nice job addressing my previous concerns with the manuscript and addressed my concerns with

Figures and data representation as well as overall clarity of bioinformatics methods used and clarification of biology where asked. The additional information and data justifying study design as well as the additional figures in the responses to my concerns was very clear and the additional references and context motivates the significance of these findings between both my and the other Review comments, I think the paper is now strong and acceptable for publication in Nature Communications.

Responses to reviewers (NCOMMS-22-05327)

Reviewer #1 (Remarks to the Author):

For the most part, the authors have addressed my main concerns, especially with respect to techniques. Although technique variations could not be ignored, as shown by the data from 293T single cells. The authors have provided alternative tool and potentially useful resources for analyzing transcription and translation during oocyte maturation. I do have further suggestions for this work.

We thank reviewer#1 for recognizing our efforts in addressing the concerns and questions.

1. The biological insights and experimental validations are the main weakness of this study. The authors included the Trim-away method, while only 30% (n=3) of the oocytes were abnormal in the extrusion of PB. So, the conclusion that OOSP2 induces human oocyte maturation is over-claimed and should be toned down. Additionally, the title needs to be revised, for example: “Single-cell transcriptome & translome dual-omics analysis of mouse and human oocytes”. This would not significantly reduce the novelty of the methodology part.

We have considered reviewer#1’s suggestion to change the title of the papers. However, after discussing with other authors, we think that TRIM-away is not the only experimental results supporting our finding that OOSP2 inducing effect of oocyte maturation. Besides TRIM-away experimental results, we have showed that recombinant OOSP2 increased the oocyte maturation rate, antibody in the culturing media inhibited the oocyte maturation, single-oocyte T&T-seq analysis showing OOSP2 upregulated many molecular pathways involved in oocyte maturation. Hence, we think that there is enough experimental evidence for us to conclude that ‘induction of human oocyte maturation by OOSP2’. Taking into account the mouse datasets we presented and suggestion by the reviewer#1, we decide to change the title to “Single-cell transcriptome & translome dual-omics analysis of mouse and human oocytes reveals induction of human oocyte maturation by OOSP2”

2. As this is a resource article, the authors need to upload all the sequenced data of this study to their GEO accessions GSE197578, and provide the reviewer link for both referees.

All the sequencing data is now available at the GEO site with accession number: GSE197578 and the source data (file size larger than 40MB) is available at:

<https://cloud.tsinghua.edu.cn/f/e019f82fdcc74aedbc50/>

3. A recently published study described a Ribo-lite method for profiling the translational landscapes of mouse oocytes and preimplantation embryos (<https://doi.org/10.1038/s41556-022-00928-6>). May the authors add the Ribo-lite method to their Table 1 and made some comments in the discussion section.

We have added the Ribo-lite method and publication to Table 1 with comments in the discussion.

Reviewer #2 (Remarks to the Author):

Hu et al. present a technology called T&T seq, referring to a dual-omics method for profiling both the transcriptome and translome in parallel, using low input material such as the oocytes profiled in this study, but it is significant to note that this technology could be applied to any cell type in that sense. The technology is also significant in that one can delineate T&T features from both the nuclear and cytoplasmic compartments for downstream functional analysis using data produced with T&Tseq. Moreover, this is clearly necessitated by the need to disambiguate gene expression and translome in whole, while most traditional technologies either fall short on the input requirement and/or need to integrate these data in-silico and/or need for targeted approaches on either the side of gene or protein expression when in parallel. This manuscript focuses on leveraging the dual-omics technology to profile the translome and transcriptomes of maturing human and mouse oocytes

from the GV to MII stage and compares those profiles to morula embryos for developmental biology purposes. In all, OOSP2 is identified, for the first time, as a key induction factor for early oocyte maturation and demonstrates the power of this method given the need for comparing gene and protein expression in the case of OOSP2, given the binding predictions and human-specific UTR CPE motifs for human versus the oosp2 mouse analog. Functional validation with OOSP2-blocked (using antibody) experiments versus control successfully shows control over oocyte maturation in developing embryos, albeit with some degree of variability.

Overall, T&Tseq is a useful method that essentially integrates existing RiboLace technology with SMARTerSeq to enable the platform, making it more of a technology advancement rather than a purely novel method, though is robustly integrated and demonstrated to enable novel findings in the OOSP2 example and also notable that competing technologies such as miniRiboSeq and RiboTag require much higher input preventing these types of single to few-cell cohorts to be profiled using existing methods without the need for in-silico data integration.

I appreciate the thorough and thoughtful responses and adjustments as suggested. The Authors did a very nice job addressing my previous concerns with the manuscript and addressed my concerns with Figures and data representation as well as overall clarity of bioinformatics methods used and clarification of biology where asked. The additional information and data justifying study design as well as the additional figures in the responses to my concerns was very clear and the additional references and context motivates the significance of these findings between both my and the other Review comments, I think the paper is now strong and acceptable for publication in Nature Communications.

We appreciate reviewer#2 for recognizing the significance of our study and our efforts in addressing the concerns and questions.